# Ensemble Distribution Distillation via Flow Matching

Jonggeon Park [* 1]   Giung Nam [* 1]   Hyunsu Kim [1]   Jongmin Yoon [1]   Juho Lee [1]

## Abstract

Neural network ensembles have proven effective in improving performance across a range of tasks; however, their high computational cost limits their applicability in resource-constrained environments or for large models. Ensemble distillation, the process of transferring knowledge from an ensemble teacher to a smaller student model, offers a promising solution to this challenge. The key is to ensure that the student model is both cost-efficient and achieves performance comparable to the ensemble teacher. With this in mind, we propose a novel ensemble distribution distillation method, which leverages flow matching to effectively transfer the diversity from the ensemble teacher to the student model. Our extensive experiments demonstrate the effectiveness of our proposed method compared to existing ensemble distillation approaches.

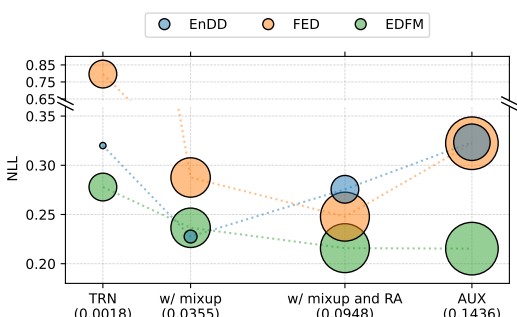

*Figure 1.* **How good is ensemble distribution distillation?** It illustrates how effectively different ensemble distribution distillation methods learn the diversity of the ensemble teacher. The x-axis represents the teacher diversity under different diversification strategies (§ 5.1), while the y-axis shows the performance of the distilled student in each setting (NLL; lower is better). The size of each circle marker indicates the level of diversity learned by the student, demonstrating how well each distillation method absorbs the teacher diversity and mimics the ensemble behavior of producing diverse predictions (§ 5.2). Our proposed EDFM approach effectively learns teacher diversity as it becomes more pronounced, continually absorbing it and outperforming baselines.

## 1. Introduction

The concept of ensembling neural networks has long been a foundational approach in machine learning (Hansen and Salamon, 1990), based on the idea of constructing a strong hypothesis by combining multiple weaker ones (Kearns, 1988). Interestingly, ensemble methods remain highly relevant in the era of deep learning, offering a simple but effective way to improve the performance of deep neural networks (Ciresan et al., 2012; Krizhevsky et al., 2012). However, despite its benefits, a notable drawback of the ensemble method becomes more pronounced with deep neural networks: *"many ensembles are large and slow"* (Buciluǎ et al., 2006). To address this, the machine learning community has naturally gravitated toward compressing the ensembles into a single, more efficient model (Buciluǎ et al., 2006; Ba and Caruana, 2014; Hinton et al., 2014).

*Ensemble distillation* is a specific form of *knowledge distillation* (Hinton et al., 2014), where the teacher model is an ensemble. Unlike the typical knowledge distillation setup, in which a single teacher model provides one prediction per training instance, ensemble distillation incorporates predictions from multiple teacher models, introducing *diversity*. While this diversity is a key distinguishing feature, simply adopting the basic knowledge distillation objective (Ba and Caruana, 2014; Hinton et al., 2014), i.e., averaging the distillation losses or teacher predictions, overlooks the diversity within the ensemble teacher. Consequently, the main goal in the ensemble distillation literature is to effectively leverage the multiple predictions, or the empirical predictive distribution, produced by the ensemble teacher (Cui et al., 2020; Malinin et al., 2020; Penso et al., 2022).

Despite recent advances, a significant performance gap remains between ensemble teachers and their distilled students. This gap arises primarily from two key limitations: *First,* deep neural networks are typically trained to achieve near-zero training errors, causing ensemble models to produce highly correlated predictions on the training set. As a

[*]Equal contribution [1]Korea Advanced Institute of Science and Technology, Republic of Korea. Correspondence to: Juho Lee <juholee@kaist.ac.kr>.

*Proceedings of the 42nd International Conference on Machine Learning*, Vancouver, Canada. PMLR 267, 2025. Copyright 2025 by the author(s).

result, ensemble distillation methods operating on the same dataset struggle to transfer meaningful diversity to student models (Nam et al., 2021). *Second,* the capacity of student models is inherently constrained. Since ensemble distillation aims to approximate ensemble predictions efficiently, there exists a tradeoff between inference speed and predictive performance, e.g., embedding multiple subnetworks within a single model (Nam et al., 2021; Penso et al., 2022) fundamentally limits the expressiveness, making it difficult to capture the full diversity of ensemble predictions.

Recently, Kim et al. (2024) reformulated ensemble distillation as a distribution-matching problem, leveraging the diffusion Schrödinger bridge (DSB) to map a single model's predictions to ensemble outputs. Unlike traditional methods, DSB enables flexible student modeling by iteratively refining predictions through a lightweight score network. However, their approach remains limited to approximating the ensemble's point estimates rather than modeling the full distribution of teacher predictions.

In this paper, we propose a novel ensemble distribution distillation algorithm based on flow matching (Lipman et al., 2023). Our approach introduces a lightweight network that learns the vector field mapping a single model's prediction to the distribution of ensemble teacher predictions. Crucially, we emphasize the role of teacher diversity in distillation and show that existing methods fail to capture this diversity effectively, whereas ours succeeds (Fig. 2). Furthermore, our method enables fast, parallelizable inference, significantly reducing wall-clock time compared to prior approaches. We substantiate our claims through extensive experiments, covering a range of tasks including image classification and language modeling. Our contributions are summarized as follows.

- We present a comprehensive analysis of ensemble distribution distillation with respect to both *diversity* and *fidelity*. In particular, we investigate when ensemble teachers exhibit diversity (§ 5.1) and how effectively distillation methods can learn from it (§§ 5.2 and 5.3).

- We propose a novel ensemble distribution distillation algorithm, Ensemble Distillation via Flow Matching (EDFM), which, as the name suggests, is based on the flow matching framework (§ 4). Our extensive experiments on image classification and language tasks validate both the efficiency (§ 5.4) and effectiveness (§§ 5.5 and 5.6) of the approach.

## 2. Preliminaries

**Notation.** We focus on a $K$-way classification problem, where a neural network parameterized by $\boldsymbol{\theta} \in \boldsymbol{\Theta}$ takes inputs $x \in \mathcal{X}$ and produces predictions for the corresponding labels $y \in \mathcal{Y}$, with *logits* $\boldsymbol{z}(x, \boldsymbol{\theta}) \in \mathbb{R}^K$. We also denote the

categorical probabilities as $\boldsymbol{p}(x, \boldsymbol{\theta}) = \text{softmax}(\boldsymbol{z}(x, \boldsymbol{\theta}))$, and their element-wise logarithm as $\log \boldsymbol{p}(x, \boldsymbol{\theta})$. For notational simplicity, we will often omit the dependence of logits and probabilties on inputs, e.g., $\boldsymbol{p}(x, \boldsymbol{\theta})$ is written as $\boldsymbol{p}_{\boldsymbol{\theta}}$ when $x$ is clear from the context.

**Ensembles.** We view ensembles within a Bayesian framework, where the model parameters $\boldsymbol{\theta}$ are treated as random variables. More precisely, the predictive categorical distribution is approximated by an ensemble of neural networks defined by the set of parameters $\{\boldsymbol{\theta}_m\}_{m=1}^M$:

$$p(y|x) = \mathbb{E}_{\boldsymbol{\theta} \sim q(\boldsymbol{\theta})}[p(y|x, \boldsymbol{\theta})] \approx \frac{1}{M} \sum_{m=1}^M p(y|x, \boldsymbol{\theta}_m), \quad (1)$$

assuming the parameters are sampled from the approximate posterior $q(\boldsymbol{\theta})$. In Bayesian literature, $q(\boldsymbol{\theta})$ is commonly modeled using a Gaussian or a mixture of Gaussians to facilitate tractable sampling (Maddox et al., 2019; Shen et al., 2024). Focusing on the classification problem, $p(y|x, \boldsymbol{\theta}) = \text{Cat}(y|\boldsymbol{p}_{\boldsymbol{\theta}}(x))$ represents a $K$-way categorical distribution with event probabilities $\boldsymbol{p}_{\boldsymbol{\theta}}(x)$.

**Ensemble distillation.** The most straightforward way to distill the ensemble teacher, defined by $q(\boldsymbol{\theta})$, into the student model, parameterized by $\boldsymbol{\phi}$, is to minimize

$$\mathbb{E}_{\boldsymbol{\theta} \sim q(\boldsymbol{\theta})} \left[ \mathcal{H} \left( \boldsymbol{p}_{\boldsymbol{\theta}}, \boldsymbol{p}_{\boldsymbol{\phi}} \right) \right], \quad (2)$$

where $\mathcal{H}[\cdot, \cdot]$ computes the cross-entropy between two categorical probabilities, and $\boldsymbol{p}_{\boldsymbol{\phi}}$ is the categorical output from the student model. Eq. 2 is the average of standard distillation losses (Hinton et al., 2014), and it is equivalent to

$$\mathcal{H} \left( \mathbb{E}_{\boldsymbol{\theta} \sim q(\boldsymbol{\theta})} \left[ \boldsymbol{p}_{\boldsymbol{\theta}} \right], \boldsymbol{p}_{\boldsymbol{\phi}} \right), \quad (3)$$

which is the standard distillation loss with the mean of the ensemble predictions. Unfortunately, this basic strategy entirely eliminates the diversity in ensemble predictions.

*Ensemble distribution distillation* approaches make the student model capture the distribution of teacher predictions. Specifically, $\boldsymbol{p} : \mathcal{X} \times \boldsymbol{\Theta} \to \Delta^{K-1}$ can be interpreted as a parametric function that maps the input space to the class-probability simplex. The random variable $\boldsymbol{\pi}_x := \boldsymbol{p}(x, \boldsymbol{\theta})$ then represents the distribution of categorical probabilities for a given $x$ under the posterior over parameters $q(\boldsymbol{\theta})$. By introducing the induced distribution $q(\boldsymbol{\pi}_x|x)$, we obtain

$$p(y|x) = \int_{\boldsymbol{\Theta}} \text{Cat}(y|\boldsymbol{p}(x, \boldsymbol{\theta})) q(\boldsymbol{\theta}) \mathrm{d}\boldsymbol{\theta}$$
$$= \int_{\Delta^{K-1}} \text{Cat}(y|\boldsymbol{\pi}_x) q(\boldsymbol{\pi}_x|x) \mathrm{d}\boldsymbol{\pi}_x. \quad (4)$$

Ultimately, the core idea of the ensemble distribution distillation scheme is to use a single student model, parameterized by $\boldsymbol{\phi}$, to model the induced distribution $q_{\boldsymbol{\phi}}(\boldsymbol{\pi}_x|x)$.

For example, a Dirichlet Prior Network (DPN; Malinin and Gales, 2018) could serve as the student model if we assume that $q(\boldsymbol{\pi}_x|x)$ follows a Dirichlet distribution. While the Dirichlet distribution is a common choice for the induced distribution (Cui et al., 2020; Malinin et al., 2020), due to its conjugacy with the categorical distribution and its ability to facilitate tractable integration in Eq. 4, it is not necessarily required. The induced distribution is defined indirectly through the observations $\{\boldsymbol{p}(x, \boldsymbol{\theta}_m)\}_{m=1}^M$, with no guarantee that they follow the Dirichlet distribution. It motivates modeling the induced distribution as arbitrary, rather than necessarily Dirichlet (Penso et al., 2022).

**Flow matching.** Flow Matching (FM; Lipman et al., 2023) is a framework for training simulation-free continuous normalizing flows (Chen et al., 2018) by learning vector fields that transport a base density $p_0(x)$ to a target density $p_1(x)$ along a deterministic probability path whose density function at time $t$ is specified as $p_t(x)$. The probability path follows the neural ordinary differential equation (ODE)

$$\mathrm{d}x_t = u_t(x_t)\mathrm{d}t, \quad x_0 \sim p_0(x_0) \tag{5}$$

where $u_t(x)$ represents the vector field that satisfies the continuity equation for mass conservation. FM is trained to minimize the discrepancy between the learned vector field $u_t$ and the ground-truth $v_t$ derived from data:

$$\mathcal{L}_{\mathrm{FM}}(\phi) = \mathbb{E}_{t,x_t}\left[\lambda(t)\left\|u_t(x_t;\phi) - v_t(x_t)\right\|^2\right], \tag{6}$$

where $t \sim \mathcal{T}$ defined over $[0,1]$, $x_t \sim p_t$, and $\lambda$ is a time-dependent weighting function. However, such a formulation Eq. 6 is challenging in general as obtaining the (marginal) ground-truth $v_t(x)$ is typically intractable. Instead, we model FM in logit space conditioned on input $x$, requiring a Conditional Flow Matching (CFM) formulation conditioned by the base density $\boldsymbol{p}_0$, where both the base and target distributions depend on an auxiliary variable $c$:

$$\begin{aligned}&\mathcal{L}_{\mathrm{CFM}}(\phi) \\ &= \mathbb{E}_{t,x_t,x_0}\left[\lambda(t)\left\|u_t(x_t|x_0,c;\phi) - v_t(x_t|x_0,c)\right\|^2\right],\end{aligned} \tag{7}$$

where $t \sim \mathcal{T}, x_0 \sim p_0(x_0|c)$, and $x_t \sim p_t(x_t|x_0, c)$. FM has been successfully applied to generative modeling due to improved training stability and computational efficiency.

## 3. Related Work

**Ensemble distillation.** In recent years, extensive research on ensemble distillation using deep neural networks has focused on three key design components: *"how to construct the ensemble teacher"* (Ba and Caruana, 2014; Korattikara et al., 2015; Bulò et al., 2016), *"how to design the student model"* (Cui et al., 2020; Malinin et al., 2020; Tran et al., 2020; Mariet et al., 2021; Penso et al., 2022), and, most importantly, *"how to distill the ensemble teacher into a single student model"* (Cui et al., 2020; Malinin et al., 2020; Du et al., 2020; Ryabinin et al., 2021; Nam et al., 2021; Nam et al., 2022; Penso et al., 2022). A central philosophy shared by most of these studies is that *"diversity matters when learning from ensembles"* (Nam et al., 2021), highlighting the importance of effectively accounting for the diversity inherent in the ensemble teacher during the distillation process. To this end, *1) one-to-one ensemble distillation* approaches design the student model to handle multiple predictions, using methods such as multi-headed modeling (Tran et al., 2020) or weight-sharing techniques (Mariet et al., 2021; Nam et al., 2021), while *2) ensemble distribution distillation* methods model the distribution of the ensemble predictions (Cui et al., 2020; Malinin et al., 2020; Ryabinin et al., 2021; Penso et al., 2022).

**Fast and efficient ensembling.** There has been a persistent demand for more cost-efficient ensemble methods, as ensembles of deep neural networks are typically associated with high computational costs during both training and inference. Research under the terms *fast* or *efficient* ensembling has aimed to address this demand, including: *"how to reduce training costs for ensemble construction"* (Huang et al., 2017; Garipov et al., 2018; Benton et al., 2021), and *"how to reduce inference costs for ensemble behaviour"* (Lee et al., 2015; Wen et al., 2020; Havasi et al., 2021; Yun et al., 2023). In particular, Yun et al. (2023) emphasized the importance of reducing test-time costs for practical applications and proposed a framework that mimics the output of one ensemble component based on the output of another using a lightweight neural network. Kim et al. (2024) further extended this approach by utilizing the Schrödinger bridge algorithm (Liu et al., 2023a), mimicking the mean of the ensemble predictions. Our approach is most closely related to these works, as it also leverages the outputs of a single ensemble component. However, the key difference is that we perform ensemble distribution distillation via flow matching, sampling from the ensemble's predictive distribution rather than producing a single point estimate, thereby better capturing the ensemble's diversity.

**Flow matching and knowledge distillation.** Meanwhile, there have also been efforts (Shao et al., 2024; Huang et al., 2023; Yao et al., 2024) to incorporate flow matching or diffusion models into knowledge distillation. These methods, however, bear greater resemblance to DBN (Kim et al., 2024) than to EDFM, as they primarily rely on one-to-one alignment of logits or intermediate features between teacher and student models. Distilling the knowledge of the diffusion model itself, especially in terms of synthesis efficiency, has also been widely investigated (Salimans and Ho, 2022; Song et al., 2023; Yin et al., 2024a; Yin et al., 2024b). We expect that these contributions might offer meaningful insights for further improvements of EDFM.

# 4. Approach

## 4.1. Flow matching in logit space

For a given input $x$, we consider the logits $\boldsymbol{z}(x, \boldsymbol{\theta})$ with $\boldsymbol{\theta} \sim q(\boldsymbol{\theta})$, predicted by the ensemble teacher with the approximate posterior $q(\boldsymbol{\theta})$. These logits are assumed to follow the distribution $\boldsymbol{p}_1$:

$$\boldsymbol{z}_1^x \sim p_1(\boldsymbol{z}_1^x | x), \qquad (8)$$

where the dependence on $\boldsymbol{\theta}$ is omitted for simplicity, and we denote $\boldsymbol{z}_1^x \equiv \boldsymbol{z}(x, \boldsymbol{\theta})$. Our goal is to construct a probability path whose density function at time $t$ is specified as $p_t$ for $t \in (0, 1)$, starting from the source distribution $p_0(\cdot|x) := \mathcal{N}(\cdot; \boldsymbol{0}, \sigma^2 \mathbf{I})$ and transitioning to the target distribution $p_1(\cdot|x)$. To achieve this, we introduce a *conditional vector field* $u : [0, 1] \times \mathbb{R}^K \times \mathcal{X} \to \mathbb{R}^K$ in the $K$-dimensional logit space:

$$u(t, \boldsymbol{z}_t^x, x) := \frac{\boldsymbol{z}_1^x - \boldsymbol{z}_t^x}{1 - t}, \qquad (9)$$

which induces a *conditional probability path*:

$$p_{t|1}(\boldsymbol{z}_t^x | \boldsymbol{z}_1^x, x) = \mathcal{N}(\boldsymbol{z}_t^x | t\boldsymbol{z}_1^x, (1-t)^2 \mathbf{I}), \qquad (10)$$

for $0 \le t \le 1$. Conditioned on $\boldsymbol{z}_1^x \sim p_1(\boldsymbol{z}_1^x | x)$, the random variables $\boldsymbol{z}_{t|1}^x$ conditioned on $t \in [0, 1]$ are given by:

$$\boldsymbol{z}_{t|1}^x = t\boldsymbol{z}_1^x + (1 - t)\boldsymbol{z}_0^x \sim p_{t|1}, \qquad (11)$$

and the marginal random variable $\boldsymbol{z}_t^x \sim p_t$ is defined as a linear combination of $\boldsymbol{z}_0^x \sim p_0$ and $\boldsymbol{z}_1^x \sim p_1$:

$$\boldsymbol{z}_t^x = t\boldsymbol{z}_1^x + (1 - t)\boldsymbol{z}_0^x \sim p_t^x. \qquad (12)$$

Building on these formulations, we model $u$ using $u_{\boldsymbol{\phi}}$, a neural network parameterized by $\boldsymbol{\phi}$. The network is trained by minimizing the conditional flow matching loss:

$$\mathcal{L}(\boldsymbol{\phi}) = \mathbb{E}_{t, \boldsymbol{z}_0^x, \boldsymbol{z}_1^x} \left[ \lambda(t) \| u_{\boldsymbol{\phi}}(t, \boldsymbol{z}_t^x, x) - (\boldsymbol{z}_1^x - \boldsymbol{z}_0^x) \|^2 \right], \qquad (13)$$

where $t \sim \mathcal{T}$, $\boldsymbol{z}_0^x \sim p_0$, $\boldsymbol{z}_1^x \sim p_1$, and $\lambda$ denotes a time-dependent weighting function (cf. § 4.3).

It is important to note that our formulation of flow matching represents one of the simplest instances, corresponding to a rectified flow (Liu et al., 2023b) between Gaussian noise and the data distribution. More sophisticated approaches are conceivable, including 1) incorporating the geometric structure of logits, such as flow matching constrained to the probability simplex, or 2) modifying the perturbation schedule, for example, adopting diffusion models instead of flow matching. Nonetheless, empirical evaluation revealed that many of these more complex formulations underperform relative to the simple approach presented here. Further ablation studies examining various formulations are provided in Appendix A.1.

## 4.2. Student network

The student network $u_{\boldsymbol{\phi}}$ is designed to condition on the input data by incorporating relevant information associated with $x$. To this end, we utilize features extracted from the penultimate layer of a pretrained teacher network—for example, representations prior to the average pooling layer in the ResNet architecture (He et al., 2016)—as conditioning inputs. Leveraging such pretrained features not only enhances the student network's ability to discriminate between different inputs but also transfers valuable knowledge embedded in the teacher network, thereby facilitating faster convergence and improved performance. Since the pretrained teacher network serving as a feature extractor remains fixed during flow matching training, the computational overhead incurred is substantially lower than that required for training the teacher network itself.

For the student network architecture, we employ a denoising multilayer perceptron (MLP) structure as proposed by Li et al. (2024). This lightweight architecture consists of a limited number of residual MLP blocks. Within each block, the perturbed sample $z_t^x$ is concatenated with the conditioning input, followed by the integration of time-embedding information via an adaptive layer normalization mechanism with zero initialization (Peebles and Xie, 2023). This processed representation then passes through an MLP layer with a skip connection. Despite being simple, this simple architecture proves sufficient to capture the target distribution, given the inherently low-dimensional nature of the logit space. Additionally, due to the computational efficiency of MLP layers on modern hardware accelerators such as GPUs, the MLP-based student network offers significant advantages in execution time, particularly for parallel sampling, as will be shown in § 5.4. It should be noted, however, that the denoising MLP is not the sole practical option for EDFM. While the MLP architecture offers a favorable balance between computational efficiency and performance, alternative architectures, such as transformers, may be more suitable in specific contexts, particularly when scalable student networks are required. Further analysis of the impact of network architecture on EDFM performance is provided in Appendix A.2.

## 4.3. Training and inference

We identified the variance of the initial Gaussian noise $\sigma$ and the time distribution $\mathcal{T}$ as critical factors influencing the performance of EDFM. Through extensive ablation studies, we set $\sigma = 4$ and defined $\mathcal{T}$ as a distribution over $[0, 1]$ that places exponentially greater weight near $t = 1$, i.e., near the data. Additionally, we adopted the preconditioning technique proposed by Karras et al. (2022) for the student network, ensuring that the network input and loss function maintain comparable scales with unit variance,

**Algorithm 1** Training EDFM

**Require:** Train dataset $\mathcal{D} = \{(x_i)\}_{i=1}^N$, noise distribution $p_0 := \mathcal{N}(\cdot; \mathbf{0}, \sigma^2 \mathbf{I})$, time distribution $\mathcal{T}$, time-dependent weighting function $\lambda$, empirical distribution of ensemble teacher logits conditioned on data $x$ $p_1(z_1^x|x)$, learning rate $\eta$, pretrained teacher network $\theta$, and student network $\phi$

1: **repeat**
2:  Sample $x \sim \mathcal{D}$
3:  Sample $z_0^x \sim p_0$, $z_1^x \sim p_1(z_1^x|x)$, $t \sim \mathcal{T}$
4:  Extract feature $\bar{x}$ of input $x$ from teacher $\theta$
5:  Obtain perturbed sample $z_t^x$ according to (12)
6:  Compute student network output $u_\phi(t, z_t^x, \bar{x})$
7:  Compute loss $\mathcal{L}_\phi$ according to (13)
8:  Update parameters: $\phi \leftarrow \phi - \eta \nabla_\phi \mathcal{L}_\phi$
9: **until** $\phi$ is converged

**Algorithm 2** Sampling from EDFM

**Require:** Test dataset $\mathcal{D} = \{(x_i)\}_{i=1}^N$, noise distribution $p_0 := \mathcal{N}(\cdot; \mathbf{0}, \sigma^2 \mathbf{I})$, number of sampling steps $N$, truncation $\epsilon$, sampling step schedule $\{1 = t_N, \ldots, t_0 = \epsilon\}$, ODE solver, pretrained teacher network $\theta$, and trained student network $\phi$

1: Sample $x \sim \mathcal{D}$
2: Sample $z_0^x \sim p_0$
3: Extract feature $\bar{x}$ of input $x$ from teacher $\theta$
4: $z \leftarrow z_0^x$
5: **for** $i = N$ to 2 in reverse **do**
6:  $z \leftarrow$ ODE solver$(z, t_i, t_{i-1}, u_\phi(t_i, z, \bar{x}))$
7: **end for**
8: $z \leftarrow z - (t_1 - t_0)u_\phi(t_1, z, \bar{x})$
9: Return $z$

while the variance of the network output is minimized. The time-dependent weighting function $\lambda$ was selected under this principle. The training algorithm for EDFM is detailed in Algorithm 1.

For sampling, the choice of ODE solver, the scheduling of sampling steps, and the total number of steps proved essential to performance. After thorough evaluation, we employed the Heun solver (Karras et al., 2022) and scheduled sampling steps to be exponentially concentrated near the data. Because our flow matching formulation reduces to rectified flow, it generates relatively straight trajectories toward the logits, facilitating efficient sampling with a reduced number of network function evaluations (NFEs). Empirically, we observed that high-quality logits can be obtained with as few as two steps (equivalent to five NFEs), even in relatively high-dimensional spaces of up to 100 dimensions. We emphasize here that the iterative nature of flow matching sampling does not substantially increase the overall sampling time. This efficiency arises because the pretrained teacher network, which serves as a fixed feature extractor, is executed only once per input, as the conditioning information remains constant throughout the sampling iterations. The algorithm for sampling from EDFM is presented in Algorithm 2. Please refer to Appendix A.3 for further details on the design choices of EDFM.

## 5. Experiments

We utilize Multi-SWAG (Wilson and Izmailov, 2020) and Multi-IVON (Shen et al., 2024) as ensemble teachers for image and language tasks, respectively, owing to their scalability and proven effectiveness (Wilson et al., 2022). For the image classification experiments in § 5.5, we use a ResNet with a depth of 32, projection shortcuts, filter response normalization, and Swish activation, as described

in Kim et al. (2024). For the commonsense reasoning experiments in § 5.6, we employ a pretrained LLaMA-2-7B base model (Touvron et al., 2023). For details on ensemble teacher construction, please see Appendices B.1 and B.2.

Unless specified, 'Mean±Std' results are reported as the mean and standard deviation calculated over three trials across tables. The best and second-best values are highlighted using **bold-faced underline** and underline, respectively. As baselines, we include KD (Hinton et al., 2014), EnDD (Ryabinin et al., 2021), FED (Penso et al., 2022), and DBN (Kim et al., 2024), and the ensemble teacher. We refer readers to Appendices B.3 and B.4 for details on evaluation metrics and baseline methods.

### 5.1. When do teacher predictions get diverse?

The existing ensemble distillation literature emphasizes the importance of ensuring sufficiently diverse teacher predictions for effective distillation (Malinin et al., 2020; Nam et al., 2021). However, deep neural networks tend to produce over-confident predictions on training data, making it difficult to capture the ensemble diversity of the teachers when this data is reused during distillation. To address this, various strategies have been employed to diversify teacher predictions. For instance, Malinin et al. (2020) employed auxiliary datasets, Nam et al. (2021) and Nam et al. (2022) developed diversifying perturbations, and Penso et al. (2022), Yun et al. (2023), and Kim et al. (2024) utilized mixup augmentation applied to the training data.

Building on these established conventions in the ensemble distillation literature, we present a comprehensive analysis of teacher diversity across various diversification strategies for image classification tasks, including: *1) RandAugment (RA; Cubuk et al., 2020)* and *2) mixup (Zhang et al., 2018)*, both of which are strong image augmentation techniques; *3) TDiv (Nam et al., 2022)*, a perturbation-based di-

Table 1. **Diversity analysis of teacher predictions.** We analyze teacher diversity by computing ensemble variance decomposition, which consists of EnsUnc, AvgUnc, and VAR. Higher VAR indicates greater diversity in ensemble predictions.

| Dataset | Split | Diversification | | | Decomposition | | |
|---|---|---|---|---|---|---|---|
| | | RA | mixup | TDiv | EnsUnc | AvgUnc | VAR (↑) |
| C10 | TRN | | | | 0.0087 | 0.0069 | 0.0018 |
| | TRN | | ✓ | | 0.0705 | 0.0350 | 0.0355 |
| | TRN | | ✓ | ✓ | 0.1212 | 0.0539 | 0.0673 |
| | TRN | ✓ | | | 0.1278 | 0.0638 | 0.0640 |
| | TRN | ✓ | ✓ | | 0.1850 | 0.0902 | 0.0948 |
| | TRN | ✓ | ✓ | ✓ | 0.1868 | 0.0853 | 0.1014 |
| | VAL | | | | 0.0845 | 0.0409 | 0.0436 |
| | AUX | | | | 0.2680 | 0.1244 | **0.1436** |

versifying approach; and *4) AUX*, which uses CINIC-10 as an auxiliary dataset. To quantify diversity without relying on ground-truth labels, we compute the ensemble variance (VAR), with higher VAR values indicating greater diversity. For detailed definitions of EnsUnc, AvgUnc, and VAR, we refer readers to Appendix B.3.

Table 1 presents the diversity of ensemble teacher predictions across the training (TRN) and validation (VAL) splits of CIFAR-10/100, as well as the auxiliary (AUX) CINIC-10 dataset. The results clearly shows that, without explicit diversification strategies, ensemble teachers produce predictions with limited diversity on training images. We hypothesize that this lack of teacher diversity can hinder effective ensemble distribution distillation when these images are reused. In § 5.2, we experimentally validate this hypothesis, demonstrating that ensemble distribution distillation improves as the teacher provides more diverse predictions through diversification strategies. In other words, a good distillation method should capture and leverage as much diversity as possible from the ensemble teacher.

## 5.2. When do ensemble distillation methods work?

In our CIFAR-10 setup, we examine the following scenarios for distilling the ensemble teacher, each capturing different levels of teacher diversity: *1) TRN*, *2) TRN w/ mixup*, *3) TRN w/ mixup and RA*, and *4) AUX*. As shown in Table 1, the diversity of the ensemble teacher gradually increases, with VAR values of [.0018, .0355, .0948, .1436]. In the AUX setup, CINIC-10 is used as an auxiliary dataset, with images from CIFAR-10 excluded (see Appendix B.1 for details), introducing a slight distribution shift (De Silva et al., 2023). We also exclude TDiv from our analysis, as its computation requires a backward pass for perturbation, which significantly increases training costs.

Fig. 1 summarizes the results of ensemble distribution distillation using EnDD, FED, and our proposed EDFM algorithms across four setups. The x-axis represents the setups

and their corresponding teacher diversity during distillation (VAR; higher values indicate greater diversity), while the y-axis shows the test performance and calibration of the distilled model (NLL and ECE; lower values indicate better performance and calibration). The size of the circle markers reflects the diversity of the distilled model on the test split (VAR; larger markers indicate greater diversity). The VAR values are computed by sampling 30 predictions for each, even for EnDD, which typically omits the sampling process when computing the mean prediction.

Fig. 1 conveys a key insight into ensemble distribution distillation: *to be where the diversity is*. As ensemble distribution distillation methods are applied to setups where teacher diversity becomes more prominent, they tend to more effectively mimic the ensemble behavior of generating diverse predictions (as reflected by the increasing circle marker sizes). However, while the student model's ability to replicate the teacher's diversity improves, this does not always lead to consistent performance gains for EnDD and FED. For EnDD, the best performance occurs in the second setup, while for FED, the highest performance is observed in the third setup. Beyond these points, despite the increased diversity, performance begins to decline.

Notably, our proposed EDFM approach not only shows higher prediction diversity but also demonstrates a consistent improvement in performance. As we will discuss in § 5.3, our method exhibits strong fidelity to the teacher; that is, the student learns not merely to generate predictions with high variance, but to genuinely replicate the ensemble teacher's behavior, thereby producing diverse predictions in a more meaningful way. To summarize, ensemble distribution distillation only can be done effectively when the teacher exhibits a certain level of diversity. However, simply learning to generate high variance is insufficient; the student must genuinely mimic the ensemble teacher's behavior, and we will validate this in our method.

## 5.3. How effective ensemble distillation methods are?

We evaluate the effectiveness of ensemble distribution distillation methods in three aspects: *1) diversity*, which measures how well the student exhibits prediction diversity, a key characteristic of the ensemble; *2) fidelity*, which assesses how closely the student mimics the teacher; and *3) generalization*, which determines whether the student can achieve performance on par with the ensemble teacher across a range of tasks. The in-depth analysis of generalization will be discussed in §§ 5.5 and 5.6, while here we focus on the diversity and fidelity analyses.

**Diversity analysis.** The operational principle of ensemble methods is *diversity*; ensembles achieve better performance when their individual members offer diverse predictions (Krogh and Vedelsby, 1994; Dieterich, 2000), a prin-

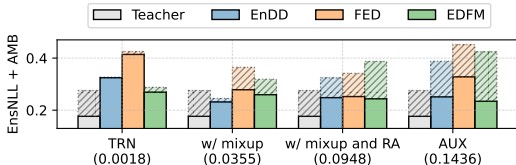

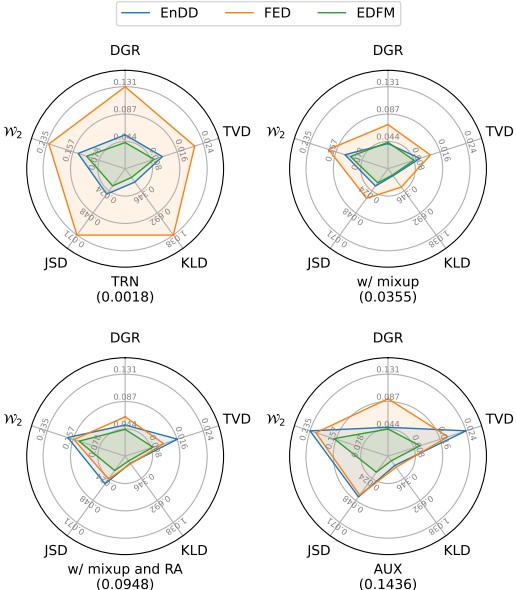

*Figure 2.* **Diversity analysis.** We compute the ambiguity decomposition to assess the diversity of the distilled model's predictions and its contribution to the ensemble gain, as shown in the shaded portion of each bar (AMB).

*Figure 3.* **Fidelity analysis.** We compute DGR, TVD, KLD, JSD, and $\mathcal{W}_2$ to assess how closely the distilled model's predictions align with those of the teacher.

ciple that holds true even in the deep learning era (Lakshmi-narayanan et al., 2017; Wilson and Izmailov, 2020; Ortega et al., 2022). To this end, we compute the ensemble ambiguity (AMB) decomposition to assess how diverse the set of predictions is and how much this diversity contributes to the ensemble gain. In particular, AMB provides insights into the performance improvement of the ensemble over the average individual performance through ambiguity decomposition, revealing the gain derived from diversity. For more details, please refer to Appendix B.3.

Fig. 2 illustrates the results of our diversity analysis on the CIFAR-10 test split. The diversity of the distilled student increase as the distillation setup more effectively captures the teacher's diversity; from (A) to (D), the shaded area, representing AMB, expands. It again highlights that ensemble distribution distillation methods successfully learn the ensemble behavior of generating diverse predictions, particularly when teacher diversity is evident; in the (A) setup, where no diversification strategies are applied, all methods fall short of the ensemble teacher's level of AMB, resulting in limited performance improvements from AMB. Notably, our EDFM not only consistently exhibits higher diversity but also shows significant gains from that diversity, as demonstrated by AMB. It clearly indicates that our method does not simply learn to generate high-variance predictions, but instead effectively harnesses diversity to drive performance improvements.

**Fidelity analysis.** Next, we analyze the effectiveness of ensemble distribution distillation methods from the perspective of *fidelity*, which quantifies how well a student model replicates the teacher's predictions. Since the core philosophy of knowledge distillation is to encourage the student to mimic the teacher (Ba and Caruana, 2014; Hinton et al., 2014), measuring the degree to which the student replicates the teacher is essential. While higher fidelity does not always guarantee better generalization—an example being the self-distillation setup of Furlanello et al. (2018), where a student that perfectly matches the teacher cannot surpass it—fidelity analysis remains crucial for understanding the underlying mechanisms, as discussed by Stanton et al. (2021). To quantify teacher-student fidelity, we compute several metrics: disagreement (AGR), total varia-

tion distance (TVD), Kullback-Leibler divergence (KLD), Jensen-Shannon divergence (JSD), and Wasserstein-2 distance ($\mathcal{W}_2$). Lower values for these metrics indicate that the student's ensemble predictions are more closely aligned with those of the ensemble teacher. For detailed definitions of each metric, we refer readers to Appendix B.3.

Fig. 3 illustrates the results of our fidelity analysis on the CIFAR-10 test split. For all the metrics we measured, our EDFM achieves higher fidelity compared to the baselines across all setups. Notably, even when comparing distributions (rather than just the mean of ensemble predictions), as seen with $\mathcal{W}_2$, EDFM demonstrates superior fidelity. It clearly indicates that our approach effectively models the induced distribution over categorical probabilities $q(\boldsymbol{\pi}_x|x)$ and closely approximates the ensemble teacher's predictive distribution $p(y|x)$ in Eq. 4.

### 5.4. How efficient ensemble distillation methods are?

To efficiently produce ensemble predictions, Nam et al. (2022) and Penso et al. (2022) use *parallelized inference* by replicating a single test input to form a mini-batch. However, unless the neural network is extremely small, processing time increases significantly as the batch size grows. On an RTX A6000, a ResNet processes a single input (batch size of 1) in 1.404 ms, while batch sizes of [32, 64, 128, 256] take [3.806, 6.993, 13.05, 24.68] ms, exhibiting a near-linear increase in our CIFAR-100 setup. It suggests inherent limits in reducing runtime when batched inference is required for ResNet, as in FED (Penso et al., 2022).

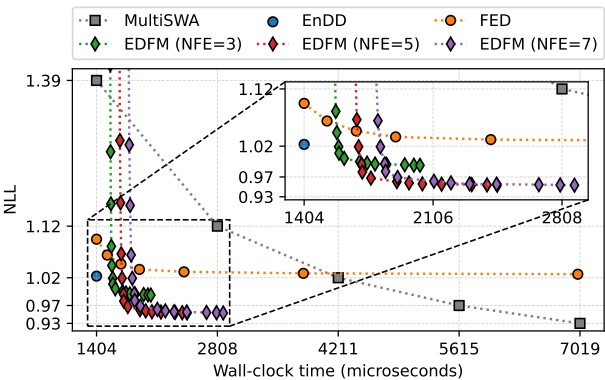

*Figure 4.* **Runtime analysis on CIFAR-100 setup.** Down and left indicate better performance and cost-efficiency, with the lower-left corner being the most desirable.

In contrast, our method performs single inference for large backbone networks (ResNet) and parallelized inference for smaller student networks (MLP), where the benefits of batch inference are more pronounced. In the same CIFAR-100 setup used for the previous analysis, processing a single input (mini-batch size of 1) with an MLP model takes 0.053 ms, while processing batched inputs with mini-batch sizes of [256, 384, 512, 640, 768, 1024] takes [0.101, 0.114, 0.128, 0.149, 0.182, 0.210] ms. Since the processing time remains at a significantly lower cost compared to ResNet, EDFM demonstrates a strong advantage in scaling ensemble predictions through parallelized inference. Building on this advantage, we investigated how the number of ensemble predictions in EDFM can scale within a constrained execution time budget.

**Runtime analysis.** Fig. 4 illustrates the evolution of both NLL and execution time as the number of student ensembles increases. While the execution time of MultiSWA ensembles scales proportionally with the number of ensembles, EDFM exhibits only a marginal increase in computational overhead. Although the execution time of FED scales at a slower rate than MultiSWA, it introduces a non-negligible overhead, and more importantly, its performance merely improves in proportion to the increased execution time. In contrast, with EDFM, the enhancement in ensemble NLL closely mirrors that of the MultiSWA teacher. This not only provides empirical evidence that EDFM effectively captures the diversity inherent in the teacher ensemble distribution, but also underscores its practical utility in enabling efficient fast ensembling.

## 5.5. Image classification tasks

We validate our proposed EDFM approach in terms of *generalization* through image classification experiments on CIFAR datasets. Since ensemble approach not only enhances in-distribution performance but also improves robustness

*Table 2.* **Evaluation results for image classification tasks.** We evaluate ACC, NLL, and ECE to assess how accurate and well-calibrated the predictions are.

| Dataset | Method | ACC (↑) | NLL (↓) | ECE (↓) |
|---|---|---|---|---|
| C10 | SWA baseline (w/o distill) | 0.932±0.001 | 0.289±0.001 | 0.046±0.000 |
| | Multi-SWAG teacher | 0.946 | 0.167 | 0.007 |
| | ⌐KD (Hinton et al., 2014) | 0.931±0.004 | 0.229±0.005 | 0.020±0.007 |
| | ⌐EnDD (Ryabinin et al., 2021) | **0.937**±0.001 | 0.228±0.004 | 0.023±0.001 |
| | ⌐DBN (Kim et al., 2024) | 0.936±0.001 | 0.218±0.002 | 0.028±0.001 |
| | ⌐FED (Penso et al., 2022) | 0.922±0.009 | 0.238±0.023 | **0.009**±0.001 |
| | ⌐EDFM (ours) | 0.931±0.000 | **0.216**±0.001 | **0.009**±0.001 |
| C100 | SWA baseline (w/o distill) | 0.735±0.004 | 1.369±0.019 | 0.173±0.004 |
| | Multi-SWAG teacher | 0.777 | 0.835 | 0.018 |
| | ⌐KD (Hinton et al., 2014) | 0.760±0.003 | 0.965±0.008 | 0.073±0.000 |
| | ⌐EnDD (Ryabinin et al., 2021) | **0.761**±0.003 | 1.031±0.002 | 0.084±0.003 |
| | ⌐DBN (Kim et al., 2024) | 0.757±0.001 | 1.002±0.004 | 0.098±0.001 |
| | ⌐FED (Penso et al., 2022) | 0.721±0.022 | 1.026±0.095 | **0.028**±0.003 |
| | ⌐EDFM (ours) | **0.761**±0.001 | **0.932**±0.003 | 0.056±0.000 |

*Table 3.* **Robustness to distribution shifts.** We evaluate ACC, NLL, and ECE to assess robustness to distribution shifts.

| Dataset | Method | ACC (↑) | NLL (↓) | ECE (↓) |
|---|---|---|---|---|
| C10.1 | SWA baseline (w/o distill) | 0.859±0.001 | 0.631±0.007 | 0.100±0.000 |
| | Multi-SWAG teacher | 0.875 | 0.386 | 0.020 |
| | ⌐KD (Hinton et al., 2014) | 0.859±0.003 | 0.480±0.027 | 0.048±0.025 |
| | ⌐EnDD (Ryabinin et al., 2021) | **0.869**±0.009 | 0.482±0.004 | 0.064±0.004 |
| | ⌐DBN (Kim et al., 2024) | 0.865±0.003 | 0.481±0.008 | 0.071±0.002 |
| | ⌐FED (Penso et al., 2022) | 0.839±0.013 | 0.481±0.047 | **0.028**±0.008 |
| | ⌐EDFM (ours) | 0.858±0.000 | **0.455**±0.000 | 0.034±0.002 |
| C10.2 | SWA baseline (w/o distill) | 0.802±0.004 | 0.990±0.004 | 0.151±0.005 |
| | Multi-SWAG teacher | 0.818 | 0.652 | 0.066 |
| | ⌐KD (Hinton et al., 2014) | 0.804±0.001 | 0.715±0.048 | 0.093±0.032 |
| | ⌐EnDD (Ryabinin et al., 2021) | **0.815**±0.005 | 0.780±0.011 | 0.112±0.004 |
| | ⌐DBN (Kim et al., 2024) | 0.808±0.001 | 0.781±0.001 | 0.118±0.001 |
| | ⌐FED (Penso et al., 2022) | 0.786±0.013 | 0.758±0.046 | **0.076**±0.011 |
| | ⌐EDFM (ours) | 0.802±0.004 | **0.666**±0.004 | **0.076**±0.002 |
| STL | SWA baseline (w/o distill) | 0.789±0.002 | 1.005±0.008 | 0.159±0.002 |
| | Multi-SWAG teacher | 0.807 | 0.615 | 0.055 |
| | ⌐KD (Hinton et al., 2014) | 0.788±0.003 | 0.726±0.060 | 0.093±0.036 |
| | ⌐EnDD (Ryabinin et al., 2021) | **0.797**±0.001 | 0.770±0.003 | 0.111±0.002 |
| | ⌐DBN (Kim et al., 2024) | 0.794±0.001 | 0.759±0.001 | 0.124±0.001 |
| | ⌐FED (Penso et al., 2022) | 0.777±0.012 | 0.698±0.040 | **0.064**±0.010 |
| | ⌐EDFM (ours) | 0.787±0.001 | **0.691**±0.006 | 0.074±0.006 |

to distribution shifts and out-of-distribution scenarios by effectively capturing predictive uncertainty (Ovadia et al., 2019), ensemble distribution distillation methods should learn these strengths from the ensemble teacher. To assess this, we evaluate our approach on CIFAR-10/100 test splits, as well as *distribution shift* datasets (CIFAR-10.1, CIFAR-10.2, and STL) and the *out-of-distribution* dataset (SVHN). For more details, please refer to Appendix B.1.

**Results.** Table 2 shows the evaluation results on CIFAR-10/100 test splits, clearly demonstrating the competitive performance of our proposed EDFM.

**Robustness to Distribution Shifts.** Table 3 presents the evaluation results on datasets associated with CIFAR-10. Since these datasets share the same 10-class labeling as CIFAR-10, we evaluate ACC, NLL, and ECE in the same manner as the standard test split. However, a distinguishing feature from the original CIFAR-10 test split is that,

*Table 4.* **Out-of-distribution detection.** We evaluate AUROC, TNR95, and TNR99 to assess out-of-distribution dection.

| Dataset | Method | AUROC (↑) | TNR95 (↑) | TNR99 (↑) |
|---|---|---|---|---|
| SVHN | SWA baseline (w/o distill) | $0.893_{\pm0.002}$ | $0.706_{\pm0.019}$ | $0.515_{\pm0.063}$ |
| | Multi-SWAG teacher | 0.935 | 0.786 | 0.679 |
| | ├KD (Hinton et al., 2014) | $0.912_{\pm0.017}$ | $0.731_{\pm0.060}$ | $0.572_{\pm0.043}$ |
| | ├EnDD (Ryabinin et al., 2021) | $0.895_{\pm0.002}$ | $0.668_{\pm0.022}$ | $0.450_{\pm0.030}$ |
| | ├DBN (Kim et al., 2024) | $0.907_{\pm0.003}$ | $0.740_{\pm0.011}$ | $0.605_{\pm0.009}$ |
| | ├FED (Penso et al., 2022) | $\underline{0.932}_{\pm0.006}$ | $\underline{0.778}_{\pm0.019}$ | $\underline{0.637}_{\pm0.052}$ |
| | └**EDFM (ours)** | $\mathbf{0.934}_{\pm0.002}$ | $\mathbf{0.784}_{\pm0.005}$ | $\mathbf{0.656}_{\pm0.035}$ |

because these datasets exhibit distribution shifts, predictions must not only be accurate but also well-calibrated. Our EDFM approach clearly not only achieves competitive ACC but also excels in NLL and ECE with wide margins, demonstrating strong robustness to distribution shifts.

**Out-of-distribution detection.** Table 4 presents the evaluation results for out-of-distribution detection, assessing the model's ability to distinguish between in-distribution and out-of-distribution inputs using receiver operating characteristic (ROC) metrics: area under the curve (AUROC) and true negative rates at 95% and 99% true positive rate (TNR95 and TNR99). We employ 1,000 in-distribution images from the CIFAR-10 test split and 1,000 out-of-distribution images from the SVHN dataset, ensuring a balanced dataset to maintain the reliability of these metrics. Our EDFM approach outperforms baselines by producing higher entropy predictions for out-of-distribution inputs, reflecting greater uncertainty.

### 5.6. Commonsense reasoning tasks

We further validate the scalability of our approach through experiments with large language models. The challenges of *large and slow* ensemble methods are amplified in modern large-scale models, where the computational burden of performing multiple forward passes through models with billions of parameters becomes particularly significant. In this context, addressing the ensemble distribution distillation problem is highly valuable; it would be more appropriate for the ensemble model to remain in the development phase, serving as a teacher for distillation rather than being directly deployed. Consequently, we assess whether our approach can effectively perform ensemble distribution distillation from a Multi-IVON teacher, which exhibits improved calibration (Shen et al., 2024; Cong et al., 2024). For experimental details on the tasks and training specifics, we refer readers to Appendix B.2.

Table 5 summarizes evaluation results on commonsense reasoning tasks, including ARC-C, ARC-E, and OBQA. The lower NLL and ECE of the Multi-IVON teacher compared to the IVON baseline, as presented in the first two rows of each group, clearly demonstrate that ensembling with samples from Multi-IVON's posterior distribution im-

*Table 5.* **Evaluation results for commonsense reasoning tasks.** We evaluate ACC, NLL, and ECE to assess how accurate and well-calibrated the predictions are.

| Dataset | Method | ACC (↑) | NLL (↓) | ECE (↓) |
|---|---|---|---|---|
| ARC-C | IVON baseline (w/o distill) | $0.710_{\pm0.011}$ | $1.874_{\pm0.167}$ | $0.246_{\pm0.009}$ |
| | Multi-IVON teacher | 0.675 | 0.891 | 0.091 |
| | ├KD (Hinton et al., 2014) | $0.662_{\pm0.032}$ | $1.126_{\pm0.080}$ | $0.218_{\pm0.015}$ |
| | └**EDFM (ours)** | $\mathbf{0.715}_{\pm0.014}$ | $\mathbf{1.113}_{\pm0.036}$ | $\mathbf{0.142}_{\pm0.011}$ |
| ARC-E | IVON baseline (w/o distill) | $0.888_{\pm0.006}$ | $0.645_{\pm0.032}$ | $0.094_{\pm0.003}$ |
| | Multi-IVON teacher | 0.868 | 0.357 | 0.026 |
| | ├KD (Hinton et al., 2014) | $0.850_{\pm0.004}$ | $0.570_{\pm0.019}$ | $0.057_{\pm0.002}$ |
| | └**EDFM (ours)** | $\mathbf{0.892}_{\pm0.002}$ | $\mathbf{0.390}_{\pm0.002}$ | $\mathbf{0.044}_{\pm0.006}$ |
| OBQA | IVON baseline (w/o distill) | $0.809_{\pm0.009}$ | $0.653_{\pm0.011}$ | $0.111_{\pm0.007}$ |
| | Multi-IVON teacher | 0.794 | 0.514 | 0.032 |
| | ├KD (Hinton et al., 2014) | $0.776_{\pm0.005}$ | $0.720_{\pm0.012}$ | $0.102_{\pm0.005}$ |
| | └**EDFM (ours)** | $\mathbf{0.818}_{\pm0.010}$ | $\mathbf{0.537}_{\pm0.005}$ | $\mathbf{0.046}_{\pm0.004}$ |

proves calibration (a slight drop in ACC is consistent with Cong et al. (2024), as we do not explicitly sharpen the posterior in our experiments). The results clearly demonstrate that our EDFM improves upon the IVON baseline by effectively distilling the Multi-IVON teacher, producing well-calibrated predictions that significantly outperform the KD baseline and closely match the teacher's calibration quality. It is worth noting that our approach still leverages a lightweight MLP architecture for flow modeling in this large language model experiment; the model consists of three blocks with a hidden dimension of 256, totaling just 1.6 million parameters—significantly smaller than the 6.7 billion parameters of the LLaMA-2 backbone and even more compact than the 4.2 million parameters required for LoRA fine-tuning. Accordingly, our approach achieves well-calibrated predictions comparable to those of the Multi-IVON teacher while incurring virtually no additional cost beyond a single LLaMA-2 forward pass—unlike the teacher, which requires multiple passes.

## 6. Conclusion

We present a novel ensemble distribution distillation approach using flow matching, where a lightweight neural network learns the vector field that transforms a single model's prediction into the predictive distribution of the ensemble teacher. Extensive experiments demonstrate that our flow matching approach effectively captures the diversity of the ensemble teacher, unlike prior works that fail to do so, which leads to substantial improvements in both accuracy and uncertainty calibration. It also enables fast, parallelizable inference, significantly reducing inference costs compared to existing distillation methods, and remains scalable even for language tasks involving modern large language models. An intriguing avenue for future research is the development of more advanced flow matching strategies specifically tailored for ensemble distillation.

## Acknowledgement

We thank Byoungwoo Park for sharing his insights. This work was partly supported by Institute of Information & communications Technology Planning & Evaluation(IITP) grant funded by the Korea government(MSIT) (No.RS-2024-00509279, Global AI Frontier Lab; No.RS-2019-II190075, Artificial Intelligence Graduate School Program(KAIST); No.RS-2022-II220713, Meta-learning Applicable to Real-world Problems; No.RS-2022-II220184, Development and Study of AI Technologies to Inexpensively Conform to Evolving Policy on Ethics). This material is based upon work supported by the Google Cloud Research Credits program with the award GCP19980904 and Cloud TPUs from Google's TPU Research Cloud (TRC).

## Impact Statement

This paper focuses on enhancing ensemble distillation frameworks with generative modeling, such as flow matching methods. By improving knowledge distillation through these approaches, our method contributes to making model compression and deployment more efficient, particularly in resource-constrained environments. Although our method is broadly applicable across various downstream tasks, it does not inherently raise ethical concerns or pose direct societal impacts. Nonetheless, as with any advancement in AI model training, the broader implications of increased efficiency and accessibility in real-world applications warrant careful consideration in future discussions.

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

# A. Ablation Study

## A.1. Flow matching formulation

Mapping Gaussian noise directly to teacher logits may initially appear overly simplistic, given that logits possess a specific geometric structure. Notably, teacher logits are invariant under scalar shifts, producing identical outputs after the softmax transformation. Flow matching formulations can be adapted to respect this geometry. Alternatively, since logits can be converted into probability vectors via the softmax operation, flow matching could be performed on the probability simplex instead of the logit space. This section explores several such approaches.

*Table 6.* Performance of various flow matching formulations on CIFAR-10.

| Space | Formulation | ACC ($\uparrow$) | NLL ($\downarrow$) | ECE ($\downarrow$) |
|---|---|---|---|---|
| Probability simplex | Statistical flow matching (Cheng et al., 2024) | 0.929 | 0.244 | 0.019 |
| | Flow matching on the probability simplex | 0.929 | 0.243 | 0.015 |
| Logit space | Flow matching with ILR | 0.931 | 0.218 | 0.011 |
| | EDM (Karras et al., 2022) | 0.931 | 0.224 | 0.009 |
| | **EDFM (Ours)** | **0.931** | **0.216** | **0.009** |

**Flow matching on the probability simplex.** Various formulations of flow matching on the probability simplex exist, such as CatFlow (Eijkelboom et al., 2024) and Dirichlet flow matching (Stark et al., 2024). However, most of these methods are inapplicable in our context as they require categorical (discrete) data, whereas we consider a continuous distribution of teacher logits defined either in logit space or equivalently on the probability simplex. Statistical flow matching (Cheng et al., 2024) is the most relevant method here, as it enables mapping from a uniform distribution to an arbitrary distribution defined on the probability simplex. Another straightforward approach is to perform basic flow matching directly on the probability simplex to transform a uniform distribution into the teacher logit distribution. This is feasible because the probability simplex is a Euclidean space where interpolations between points remain within the simplex.

Despite the theoretical appeal, empirical results shown in Table 6 demonstrate that flow matching on the simplex yields inferior performance. We attribute this to the requirement that the student network match the teacher distribution with high precision on the simplex, whereas in logit space, minor mismatches are attenuated by the subsequent softmax operation.

**Isometric log-ratio (ILR) transform.** Given the shortcomings of operating directly on the probability simplex, one may consider improvements in the logit space. Since teacher logits are invariant under scalar shifts (e.g., logits $\{-1.5, -2, 2\}$ are equivalent to $\{-0.5, -1, 3\}$ post-softmax), enforcing exact matching of logits may be overly restrictive. The isometric log-ratio (ILR) transformation from compositional data analysis offers a principled way to address this issue: it maps the $D$-dimensional probability simplex to a $(D - 1)$-dimensional Euclidean space. By applying ILR to teacher logits, flow matching can be trained in $\mathbb{R}^{D-1}$. Nonetheless, empirical results (Table 6) indicate negligible differences compared to the baseline, likely because the dimensionality reduction by one does not substantially affect the model's expressiveness in relatively high-dimensional settings.

**Diffusion models.** Aside from flow matching, diffusion models present an alternative framework to model the teacher logit distribution. Exploring such generative approaches for ensemble distribution distillation constitutes a promising future direction. We implemented the EDM model (Karras et al., 2022) to train the student network and compared it to our EDFM method. On the CIFAR-10 dataset, EDM achieved an NLL of 0.224 using 35 NFEs, whereas EDFM attained a better NLL of 0.216 with only 5 NFEs (Table 6). Notably, EDM's performance deteriorated significantly with 5 NFEs, achieving an NLL of 0.370, underscoring the superior efficiency and effectiveness of EDFM.

In summary, applying flow matching directly to pre-softmax logits yields superior results compared to modeling on post-softmax categorical probabilities. This improvement may be explained by the distortion introduced by the softmax transformation, which can obscure important scale information intrinsic to the logits. For example, the unnormalized density, given by the log-sum-exp of logits (Duvenaud et al., 2020), may be lost after softmax, thereby negatively impacting ensemble distribution distillation.

## A.2. Network architecture of student model

We performed an ablation study to evaluate the impact of network architectures of varying scales on performance. The architectures considered include 1) the denoising MLP (Li et al., 2024) already employed in EDFM, 2) the U-Net (Ronneberger et al., 2015), which is commonly used in flow matching for image synthesis and was also previously adopted in DBN (Kim et al., 2024), and 3) Transformer (Vaswani et al., 2017) models.

Table 7. Performance and efficiency of EDFM with various network architectures on CIFAR-10.

| Architecture | # of Param. | Time ($\mu s$) | ACC ($\uparrow$) | NLL ($\downarrow$) | ECE ($\downarrow$) |
|---|---|---|---|---|---|
| MLP | 0.14M | 39 | 0.929 | 0.229 | 0.013 |
| MLP (main text) | 0.33M | 68 | 0.931 | 0.216 | 0.009 |
| MLP | 0.70M | 132 | 0.930 | 0.218 | 0.011 |
| Transformer | 0.36M | 204 | 0.930 | 0.218 | 0.011 |
| Transformer | 1.31M | 724 | 0.931 | 0.213 | 0.007 |
| U-Net | 0.44M | 653 | 0.929 | 0.222 | 0.012 |

The results, summarized in Table 7, reveal that larger Transformers (with 1.31 million parameters) achieve improvements in NLL and ECE metrics but are not cost-effective, as reflected by the number of parameters and wall-clock time, where the latter denotes the time required to generate 256 ensemble predictions via batched inference. In contrast, the U-Net architecture underperforms relative to both the MLP and Transformer, which is likely attributable to its two-dimensional spatial processing being ill-suited for modeling logits. These findings suggest that the MLP architecture strikes a favorable balance, delivering sufficient accuracy for ensemble distribution distillation while offering a significant latency advantage.

## A.3. Design choice of EDFM

In this section, we elaborate on specific design choices of EDFM presented in Algorithms 1 and 2.

**Time distribution $\mathcal{T}$.** We swept over uniform, beta, and exponential distributions to find the most suitable $\mathcal{T}$, where the probability density functions are given in Table 8. The support of $\mathcal{T}$ is set to $[\epsilon, 1]$ instead of $[0, 1]$ to avoid numerical instability.

Table 8. PDFs of distributions.

| Distribution | PDF $f_T(t)$ |
|---|---|
| Uniform | $f_T(t) = 1$ |
| Beta | $f_T(t; \alpha, \beta) = \Gamma(\alpha + \beta)/(\Gamma(\alpha)\Gamma(\beta))t^{\alpha-1}(1-t)^{\beta-1}$ |
| Exponential | $f_T(t; \alpha) = \alpha^t \log \alpha/(\alpha^{1-\epsilon} - 1)$ |

The PDF of the exponential distribution is slightly modified from the conventional one to account for the support $[\epsilon, 1]$ and an arbitrary base. The choice of beta and exponential distributions was inspired by the well-known lesson in the field of diffusion models (Karras et al., 2022) that the network should learn more near the data. Specifically, the exponential distribution allocates exponentially more samples as $t$ grows, while the beta distribution achieves a peak somewhere between $t = 0$ and $t = 1$. We assigned the peak to be near the data by setting $\alpha = 5$ and $\beta = 2$. For the exponential distribution, we set $\alpha = 3$. The truncation $\epsilon$ was set to 0.001.

Experiment results shown in Table 9 indicate that the beta distribution achieved better results in terms of ECE, while performing on par with the exponential distribution on other metrics. This motivates further explorations for even better options, possibly inspired by recent progress in flow matching (Lee et al., 2024; Kim et al., 2025). We leave this as future work.

*Table 9.* Ablation study on time distribution.

| Dataset | Time distribution | ACC ($\uparrow$) | NLL ($\downarrow$) | ECE ($\downarrow$) |
|---------|------------------|-----|-----|-----|
| C10 | Uniform | 0.931 | 0.219 | 0.011 |
| | Beta | 0.931 | 0.216 | 0.008 |
| | Exp. (Ours) | 0.931 | 0.216 | 0.009 |
| C100 | Uniform | 0.761 | 0.966 | 0.066 |
| | Beta | 0.760 | 0.932 | 0.039 |
| | Exp. (Ours) | 0.761 | 0.932 | 0.056 |

**Time-dependent weighting function $\lambda$.** The time-dependent weighting function $\lambda$ was determined according to the preconditioning technique introduced in EDM (Karras et al., 2022). Specifically, in EDM, the authors first let the neural network predict the denoiser $h(\mathbf{x}_t, t)$, namely $\mathbf{x}$-prediction, and controlled the scale of the network input, output, and the weighting function of the loss so that the network input and the loss function maintain unit variance, and the variance of the network output is minimized.

*Table 10.* Comparing between $\mathbf{x}$ and $\mathbf{v}$ predictions for EDFM.

| Dataset | Prediction type | ACC ($\uparrow$) | NLL ($\downarrow$) | ECE ($\downarrow$) |
|---------|-----------------|-----|-----|-----|
| C10 | $\mathbf{x}$-pred | 0.931 | 0.234 | 0.024 |
| | $\mathbf{v}$-pred (Ours) | 0.931 | 0.216 | 0.009 |
| C100 | $\mathbf{x}$-pred | 0.760 | 1.031 | 0.090 |
| | $\mathbf{v}$-pred (Ours) | 0.761 | 0.932 | 0.056 |

However, as shown in Table 10, we found that directly predicting the velocity, namely $\mathbf{v}$-prediction, performs better than $\mathbf{x}$-prediction for EDFM. Thus, although we followed the principles suggested in EDM, our preconditioning slightly deviates from the original.

For completeness, we present the derivation for our preconditioning. The student network accepts three inputs: perturbed sample $\mathbf{z}_t^x$, time $t$, and embedding from the teacher network $\text{Emb}(x)$. Two of these three inputs are rescaled to $c_{\text{in}}(t)\mathbf{z}_t^x$, $c_{\text{time}}(t)$ before being fed into the network. The embedding $\text{Emb}(x)$ is left invariant as it already maintains stable variance, after being processed by the teacher network.

We set $c_{\text{time}}(t) = \log\left(1000(1-t)+\epsilon\right)/4$, where $\epsilon$ is a small number $10^{-12}$ to avoid numerical instability. This was inspired by the original formulation of $\log t/4$, and the transformation $t \to 1000(1-t)$ was introduced to match the scales.

Then we added a skip connection to the output of the student network $F_\phi$:

$$D_\phi(\mathbf{z}_t^x) = c_{\text{skip}}(t)\mathbf{z}_t^x + c_{\text{out}}(t)F_\phi(c_{\text{in}}(t)\mathbf{z}_t^x) \tag{14}$$

which is then used to compute the loss function

$$\mathbb{E}_{t,\mathbf{z}_0^x,\mathbf{z}_1^x}\left[\lambda(t)\|D_\phi(z_t^x) - (\mathbf{z}_1^x - \mathbf{z}_0^x)\|^2\right]. \tag{15}$$

Rearranging the terms results in

$$\mathbb{E}_{t,\mathbf{z}_0^x,\mathbf{z}_1^x}\left[\frac{\lambda(t)}{c_{\text{out}}(t)^2}\|(tc_{\text{skip}}(t)-1)\mathbf{z}_1^x + ((1-t)c_{\text{skip}}(t)+1)\mathbf{z}_0^x + F_\phi(c_{\text{in}}(t)\mathbf{z}_t^x)\|^2\right]. \tag{16}$$

We want the network input, and loss function to maintain unit variance across different time $t$, while the variance of the

network output is minimized. This means that

$$\text{Var}_{\boldsymbol{z}_0^x, \boldsymbol{z}_1^x}[c_{\text{in}}(t)\boldsymbol{z}_t^x] = \text{Var}_{\boldsymbol{z}_0^x, \boldsymbol{z}_1^x}[c_{\text{in}}(t)(t\boldsymbol{z}_1^x + (1-t)\boldsymbol{z}_0^x)] = 1 \tag{17}$$

$$c_{\text{skip}}(t) = \underset{c_{\text{skip}}(t)}{\arg\min} \text{Var}_{\boldsymbol{z}_0^x, \boldsymbol{z}_1^x}[(tc_{\text{skip}}(t) - 1)\boldsymbol{z}_1^x + ((1-t)c_{\text{skip}}(t) + 1)\boldsymbol{z}_0^x] \tag{18}$$

$$c_{\text{out}}(t) = \text{Var}_{\boldsymbol{z}_0^x, \boldsymbol{z}_1^x}[(tc_{\text{skip}}(t) - 1)\boldsymbol{z}_1^x + ((1-t)c_{\text{skip}}(t) + 1)\boldsymbol{z}_0^x] \tag{19}$$

$$\lambda(t)/c_{\text{out}}(t)^2 = 1 \tag{20}$$

Note that as $\boldsymbol{z}_0^x \sim \mathcal{N}(\cdot; \boldsymbol{0}, \sigma^2\mathbf{I})$, $\text{Var}[\boldsymbol{z}_0^x] = \sigma^2$. Also, we let $\text{Var}[\boldsymbol{z}_1^x] = \sigma_{\text{data}}^2$. Evaluating for $c_{\text{in}}(t), c_{\text{out}}(t), c_{\text{skip}}(t), \lambda(t)$ under this principle yields

$$c_{\text{in}}(t) = \frac{1}{\sqrt{t^2\sigma_{\text{data}}^2 + (1-t)^2\sigma^2}} \tag{21}$$

$$c_{\text{out}}(t) = \frac{\sigma\sigma_{\text{data}}}{\sqrt{t^2\sigma_{\text{data}}^2 + (1-t)^2\sigma^2}} \tag{22}$$

$$c_{\text{skip}}(t) = \frac{t\sigma_{\text{data}}^2 - (1-t)\sigma^2}{t^2\sigma_{\text{data}}^2 + (1-t)^2\sigma^2} \tag{23}$$

$$\lambda(t) = \frac{\sigma^2\sigma_{\text{data}}^2}{t^2\sigma_{\text{data}}^2 + (1-t)^2\sigma^2} \tag{24}$$

**ODE solver and sampling step scheduling** For ODE solver and sampling step scheduling, we ablated between Euler, Heun solvers (Karras et al., 2022) and uniform, exponential scheduling. For $N$ steps of sampling, the sampling steps are $\{i/N\}_{i=0}^N$ for uniform scheduling and $\{(1-\alpha^i)/(1-\alpha^N)\}_{i=0}^N$ for exponential scheduling. For exponential scheduling, we set $\alpha = 0.7$ for CIFAR-10 and $\alpha = 0.5$ for CIFAR-100. As Fig. 5 indicate, Heun solvers with exponential scheduling achieved the best NLL at 5 sampling steps, which amounts to 7 NFEs for Heun solver. All experiment results presented in the main paper were obtained under this configuration.

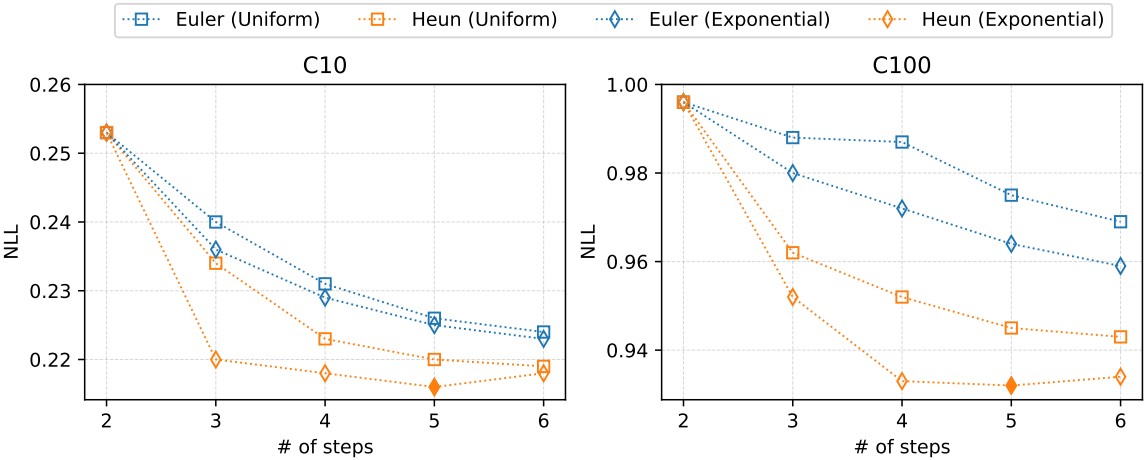

*Figure 5.* Ablation study on ODE solver and sampling step scheduling.

# B. Experimental details

## B.1. Image classification tasks

**Datasets.** Our experiments on image classification tasks utilize the following datasets:

- **CIFAR-10/100** (Krizhevsky and Hinton, 2009): It consists of 40,960 training images, 9,040 validation images, and 10,000 test images, each with a resolution of $32 \times 32 \times 3$ across 10/100 classes. As there is no officially predefined validation split, we manually partitioned the 50,000 training examples into 40,960 for training and 9,040 for validation. The datasets are publicly available at http://www.cs.toronto.edu/~kriz/cifar.html under unspecified license.

- **CINIC-10** (Darlow et al., 2018): It serves as an auxilairy train split consisting of 210,000 images with a resolution of $32 \times 32 \times 3$. The dataset is publicly available at https://github.com/BayesWatch/cinic-10 under MIT license. The original dataset contains of a total of 270,000 images, from which we dropped 60,000 CIFAR-originated images to avoid dataset leakge in our CIFAR-10/100 experiments.

- **CIFAR-10.1** (Recht et al., 2018): It serves as a test split consisting of 2,000 images with a resolution of $32 \times 32 \times 3$. The dataset is publicly available at https://github.com/modestyachts/CIFAR-10.1 under MIT license.

- **CIFAR-10.2** (Lu et al., 2020): It serves as a test split consisting of 2,000 images with a resolution of $32 \times 32 \times 3$. The dataset is publicly available at https://github.com/modestyachts/cifar-10.2 under unspecified license.

- **STL** (Coates et al., 2011): It serves as a test split consisting of 7,200 images with a resolution of $32 \times 32 \times 3$. The dataset is publicly available at https://cs.stanford.edu/~acoates/stl10/ under unspecified license. The original test split consists of 8,000 images with a resolution of $96 \times 96 \times 3$, which we resized to $32 \times 32 \times 3$. We excluded the 'monkey' class, as it is not present in CIFAR, removing 800 images in the process. Furthermore, the 'car' class was mapped to the 'automobile' class to match CIFAR's class definitions.

- **SVHN** (Netzer et al., 2011): It serves as an out-of-distribution split consisting of $73,257$ training and $26,032$ test images with colored number digit (0-9) with a resolution of $32 \times 32 \times 3$ The dataset is publicly available at http://ufldl.stanford.edu/housenumbers/ under unspecified license.

**Optimization details.** We provide details on the construction of the ensemble teacher and the training procedure for distilled students in the image classification experiments.

- **Multi-SWAG teacher**: Multi-SWAG teachers were obtained by adding diagonal covariances to each of 10 different SWA models. Each SWA model was pretrained for 800 epochs, followed by 200 epochs of SWA training with frequency 1. Using SGD optimizer with momentum 0.9, cosine decay schedule was applied for pretraining, in which the learning rate evolves from 0.1 to 0.01 and remains constant of 0.01 whilst SWA training. Batch size is 256.

- **KD**: KD was trained using SGD optimizer with momentum 0.9, learning rate 0.1 with cosine decay schedule, 1000 epochs, batch size 256. Unlike CIFAR-10, for models trained with CIFAR-100, learning rate was warmed up from 0 to 0.1 for the first 5 steps. Number of teachers to produce the target prediction is 30.

- **EnDD**: EnDD was trained with almost identical configuration with that of KD, except that the learning rate is decreased to 0.05.

- **FED**: For FED, we strictly followed the configurations provided in the original paper (Penso et al., 2022), except that we increased the number of epochs from 200 to 250 for coherence with other methods. Training further than 250 epochs showed no improvements in performance.

- **DBN**: Like FED, we strictly followed the configurations provided in the original paper (Kim et al., 2024), except that we increased the number of epochs from 800 to 1000 for coherence with other methods.

- **EDFM**: We used the denoising MLP architecture with four blocks and a hidden dimension of 256 for CIFAR-10 and 512 for CIFAR-100. The model employs a student network distilled with KD using the MultiSWAG teachers as its backbone and takes its output embedding as input. Evaluation was conducted with seven NFEs, and training was performed using the SGD optimizer with a batch size of 256, momentum of 0.9, weight decay of 5e-04, learning rates of 1e-04 for CIFAR-10 and 3e-04 for CIFAR-100, and a cosine decay learning rate schedule over 1,000 epochs.

## B.2. Commonsense reasoning tasks

**Datasets.** Our experiments on commonsense reasoning tasks build upon the work of Cong et al. (2024). Following their training and evaluation recipes, all datasets are composed exclusively of multiple-choice questions with four options:

- **ARC-Challenge (ARC-C; Clark et al., 2018)**: It consists of 1,117 questions for training and 295 questions for evaluation, publicly available at https://huggingface.co/datasets/allenai/ai2_arc under CC-BY-SA-4.0 license.

- **ARC-Easy (ARC-E; Clark et al., 2018)**: It consists of 2,241 questions for training and 567 questions for evaluation, publicly available at https://huggingface.co/datasets/allenai/ai2_arc under CC-BY-SA-4.0 license.

- **OpenBookQA (OBQA; Mihaylov et al., 2018)**: It consists of 4,957 questions for training and 500 questions for evaluation, publicly available at https://huggingface.co/datasets/allenai/openbookqa under unspecified license.

**Optimization details.** We provide details on the construction of the ensemble teacher and the training procedure for distilled students in the commonsense reasoning experiments.

- **Multi-IVON teacher**: We obtained the Multi-IVON teacher using the official codebase from Cong et al. (2024), which is available at https://github.com/team-approx-bayes/ivon-lora under unspecified license. More specifically, it involves performing LoRA fine-tuning on the pretrained LLaMA-2-7B model, which is publicly available at https://huggingface.co/meta-llama/Llama-2-7b under the Meta license[1], using the IVON optimizer (Shen et al., 2024). Using their configurations, we trained 10 IVON models with different random seeds and constructed the Multi-IVON teacher, which generates a total of 100 logits by extracting 10 logits from each individual model.

- **KD**: We fine-tuned the pretrained LLaMA-2-7B weights using the KD objective, as defined in Eq. 37, with the Multi-IVON teacher. After several attempts, we found that effective distillation requires not only applying LoRA to the query and value weights of the attention layers but also training the classification head. As a result, the number of trainable parameters is 4,194,304, which is 0.062% of the total 6,742,609,920 parameters. Moreover, the KD objective alone was insufficient for stable learning in our commonsense reasoning experiments. Therefore, as in Hinton et al. (2014), we combined the KD objective with a cross-entropy loss using ground-truth labels, weighted at 0.5. The fine-tuning was carried out using the Adam optimizer with a batch size of 4, a maximum sequence length of 320, a learning rate of 3e-05, and a linear decay learning rate schedule over 50,000 steps.

- **EDFM**: For flow modeling, we used the denoising MLP architecture with three blocks and a hidden dimension of 256, totaling 1,639,056 parameters. The model employs one ensemble component of the Multi-IVON teacher as its backbone and takes its output embedding as input. Evaluation was conducted with three NFEs, and training was performed using the SGD optimizer with a batch size of 256, momentum of 0.9, a learning rate of 1e-04, and a cosine decay learning rate schedule over 50,000 steps.

## B.3. Evaluation metrics

**Classification accuracy (ACC)**, is a metric used to evaluate the performance of a classification model by computing the proportion of correct predictions. Let $y_n$ be the true label, $\boldsymbol{p}_n^{(k)}$ the predicted probability for class $k$ for the $n^{\text{th}}$ sample, and $N$ the total number of samples to be evaluated. We then compute ACC as follows, where $[\cdot]$ denotes the Iverson bracket:

$$\text{ACC} = \frac{1}{N} \sum_{n=1}^{N} \left[ y_n = \arg\max_k \boldsymbol{p}_n^{(k)} \right]. \tag{25}$$

**Negative log-likelihood (NLL)**, also known as cross-entropy loss in classification tasks, is another important metric used for both training and evaluating classification models. It measures how well the predicted probability distribution matches the true labels. Let $y_n$ be the true label, $\boldsymbol{p}_n^{(k)}$ the predicted probability for class $k$ for the $n^{\text{th}}$ sample, and $N$ the total number of samples to be evaluated. We then compute NLL as follows:

$$\text{NLL} = -\frac{1}{N} \sum_{n=1}^{N} \log \boldsymbol{p}_n^{(y_n)}. \tag{26}$$

---

[1] https://ai.meta.com/llama/license/

**Expected calibration error (ECE;** Pakdaman Naeini et al., 2015**),** is a metric that evaluates how well a predicted probabilities correspond to the true likelihood of correct predictions; a well-calibrated model outputs probabilities that closely align with the observed frequency of correctness. Let $y_n$ be the true label, $\boldsymbol{p}_n^{(k)}$ the predicted probability for class $k$ for the $n^{\text{th}}$ sample, and $N$ the total number of samples to be evaluated. The samples are partitioned into $B$ bins based on the maximum predicted probability, $\max_k \boldsymbol{p}_n^{(k)}$, where $\mathcal{B}_b$ and $|\mathcal{B}_b|$ denote the indices and count of samples in the $b^{\text{th}}$ bin, respectively. For $b = 1, ..., B$, the observed accuracy $\text{acc}_b$ and the average confidence $\text{conf}_b$ are computed as follows:

$$\text{acc}_b = \frac{1}{|\mathcal{B}_b|} \sum_{n \in \mathcal{B}_b} \left[ y_n = \arg\max_k \boldsymbol{p}_n^{(k)} \right] \text{ and } \text{conf}_b = \frac{1}{|\mathcal{B}_b|} \sum_{n \in \mathcal{B}_b} \max_k \boldsymbol{p}_n^{(k)}. \tag{27}$$

We then compute ECE as follows:

$$\text{ECE} = \sum_{b=1}^{B} \frac{|\mathcal{B}_b|}{N} \cdot |\text{acc}_b - \text{conf}_b|. \tag{28}$$

**Ensemble ambiguity (AMB),** is a metric used to measure the diversity of ensemble predictions. Let $y_n$ be the true label, $\boldsymbol{z}_{m,n}^{(k)}$ and $\boldsymbol{p}_{m,n}^{(k)}$ the logit and categorical probability for class $k$ predicted by the $m^{\text{th}}$ ensemble member for the $n^{\text{th}}$ sample, $M$ the ensemble size, and $N$ the total number of samples to be evaluated. We then compute AMB as follows:

$$\text{AMB} = \frac{1}{NM} \sum_{n=1}^{N} \sum_{m=1}^{M} \sum_{k=1}^{K} \bar{\boldsymbol{p}}_n^{(k)} \log \frac{\bar{\boldsymbol{p}}_n^{(k)}}{\boldsymbol{p}_{m,n}^{(k)}}, \tag{29}$$

where $\bar{\boldsymbol{p}}$ denotes a normalized geometric mean of categorical probability here,

$$\bar{\boldsymbol{p}}_n^{(k)} = \frac{\bar{\boldsymbol{q}}_n^{(k)}}{\sum_{j=1}^{K} \bar{\boldsymbol{q}}_n^{(j)}} \text{ with } \bar{\boldsymbol{q}}_n^{(k)} = \prod_{m=1}^{M} \left( \boldsymbol{p}_{m,n}^{(k)} \right)^{1/M}, \text{ for } k = 1, ..., K. \tag{30}$$

We also have the following generalized ambiguity decomposition (Wood et al., 2023), which suggests that to minimize ensemble loss, ensemble ambiguity should increase while average loss decreases:

$$\underbrace{-\frac{1}{N} \sum_{n=1}^{N} \log \text{softmax}^{(y_n)} \left( \frac{1}{M} \sum_{m=1}^{M} \boldsymbol{z}_{m,n} \right)}_{\text{ensemble loss (EnsLoss)}} = \underbrace{-\frac{1}{M} \sum_{m=1}^{M} \frac{1}{N} \sum_{n=1}^{N} \log \text{softmax}^{(y_n)}(\boldsymbol{z}_{m,n})}_{\text{average loss (AvgLoss)}} - \text{AMB}. \tag{31}$$

In Fig. 2, we compute the ambiguity decomposition after applying temperature scaling (Guo et al., 2017), using the optimal temperature obtained by minimizing EnsNLL. This approach offers a more accurate comparison, as EnsNLL uses ensembled logits instead of ensembled probabilities. When a logit shift occurs for each ensemble member, the probability ensemble remains unchanged, while the logit ensemble can vary.

**Ensemble variance (VAR),** is another metric used to measure the diversity of ensemble predictions. Let $\boldsymbol{p}_{m,n}^{(k)}$ be the categorical probability for class $k$ predicted by the $m^{\text{th}}$ ensemble member for the $n^{\text{th}}$ sample, $M$ the ensemble size, and $N$ the total number of samples to be evaluated. We then compute VAR as follows:

$$\text{VAR} = \frac{1}{N} \sum_{n=1}^{N} \sum_{k=1}^{K} \underset{m=1,...,M}{\text{Variance}} \left[ \boldsymbol{p}_{m,n}^{(k)} \right] \tag{32}$$

We also have the following uncertainty decomposition (Abe et al., 2022), which suggests that the variance quantifies the extent to which predictive uncertainty increases after ensembling:

$$\underbrace{\frac{1}{N} \sum_{n=1}^{N} \left[ 1 - \sum_{k=1}^{K} \left( \bar{\boldsymbol{p}}_n^{(k)} \right)^2 \right]}_{\text{ensemble uncertainty (EnsUnc)}} = \underbrace{\frac{1}{NM} \sum_{n=1}^{N} \sum_{m=1}^{M} \left[ 1 - \sum_{k=1}^{K} \left( \boldsymbol{p}_{m,n}^{(k)} \right)^2 \right]}_{\text{average uncertainty (AvgUnc)}} + \text{VAR}. \tag{33}$$

**Agreement (AGR)**, is a commonly used metric in classification tasks to measure how often two predictions match each other. Let $\boldsymbol{p}_n^{(k)}$ and $\boldsymbol{q}_n^{(k)}$ be the two categorical probabilities for class $k$ predicted for the $n^{\text{th}}$ sample, and $N$ the total number of samples to be evaluated. We then compute AGR as follows:

$$\text{AGR} = \frac{1}{N} \sum_{n=1}^{N} \left[ \arg\max_k \boldsymbol{p}_n^{(k)} = \arg\max_k \boldsymbol{q}_n^{(k)} \right] \tag{34}$$

**Total variation distance (TVD)**, is a metric that measures the difference between two probability distributions. It quantifies the maximum discrepancy in probability mass between them. $\boldsymbol{p}_n^{(k)}$ and $\boldsymbol{q}_n^{(k)}$ be the two categorical probabilities for class $k$ predicted for the $n^{\text{th}}$ sample, and $N$ the total number of samples to be evaluated. We then compute TVD as follows:

$$\text{TVD} = \frac{1}{2N} \sum_{n=1}^{N} \sum_{k=1}^{K} \left| \boldsymbol{p}_n^{(k)} - \boldsymbol{q}_n^{(k)} \right| \tag{35}$$

**Wasserstein-2 distance** ($\mathcal{W}_2$), measures the distance between two probability distributions by quantifying the optimal cost of transporting one distribution to another. Fréchet Inception Distance (FID) (Heusel et al., 2017), a widely used metric in generative models literature, is also a specific instance of $\mathcal{W}_2$. Inspired by this, we also evaluated the fidelity of generated logits to teacher logits by measuring Wasserstein-2 distance between the two distributions. Let $P$ and $Q$ be two discrete probability distributions with support $\{x_1, \ldots, x_n\}$ and $\{y_1, \ldots, y_m\}$ respectively, so that $P = \sum_{i=1}^{n} p_i \delta_{x_i}$ and $Q = \sum_{j=1}^{m} q_j \delta_{y_j}$ where $p_i$, $q_j$ are probability masses and $\delta_x$ Dirac delta function. We then compute Wasserstein-2 distance between two discrete distributions as:

$$\mathcal{W}_2(P, Q) = \left( \inf_{\gamma \in \Gamma(P,Q)} \sum_{i=1}^{n} \sum_{i=1}^{m} \gamma_{i,j} \|x_i - y_j\|^2 \right)^{\frac{1}{2}} \tag{36}$$

where $\gamma_{i,j}$ is the transport plan meaning how much mass from $p_i$ to $q_j$, $\Gamma(P,Q)$ is the set of all possible valid transport plans that satisfy the marginal constraint $\sum_j \gamma_{i,j} = p_i, \sum_i \gamma_{i,j} = q_i$. Evaluating this involves linear programming, and can efficiently be solved using Hungarian algorithm. Note that $\mathcal{W}_2$ distance differs from TVD, in that it measures the similarity between the whole predictive distributions, where as TVD only compares their means.

### B.4. Ensemble distillation methods

**Knowledge distillation (KD; Hinton et al., 2014)** is considered the simplest method for transferring knowledge from a large and complex teacher model to a smaller student model. The training objective for knowledge distillation is given as follows:

$$\mathcal{L}_{\text{KD}}(\boldsymbol{\phi}) = \mathcal{H}\left( \bar{\boldsymbol{p}}, \boldsymbol{q}_{\boldsymbol{\phi}} \right) = \frac{1}{M} \sum_{m=1}^{M} \mathcal{H}(\boldsymbol{p}_m, \boldsymbol{q}_{\boldsymbol{\phi}}), \tag{37}$$

which simply is the cross-entropy loss widely used in classification tasks. One thing that differs is, that it is typical to soften (or harden) the teacher logits by dividing them with a temperature $T$, for better performance.

**Ensemble distribution distillation (EnDD; Malinin et al., 2020; Ryabinin et al., 2021)** assumes that the categorical predictions from the ensemble teacher follow a Dirichlet distribution. In other words, the induced distribution in Eq. 4 is assumed to be Dirichlet, i.e., $\boldsymbol{p}_1, ..., \boldsymbol{p}_M \sim q(\boldsymbol{\pi}_x | x) = \text{Dir}(\boldsymbol{\beta})$, where the concentration parameter $\boldsymbol{\beta}$ is estimated in closed-form using the approximate maximum likelihood procedure (Minka, 2000):

$$\boldsymbol{\beta}^{(k)} \approx \bar{\boldsymbol{p}}^{(k)} \cdot \frac{K-1}{2} \cdot \frac{1}{\sum_{j=1}^{K} \bar{\boldsymbol{p}}^{(j)} \cdot (\log \bar{\boldsymbol{p}}^{(j)} - \overline{\log \boldsymbol{p}}^{(j)})} \quad \text{for } k = 1, ..., K, \tag{38}$$

where $\bar{\boldsymbol{p}} = \frac{1}{M} \sum_{m=1}^{M} \boldsymbol{p}_m$ and $\overline{\log \boldsymbol{p}} = \frac{1}{M} \sum_{m=1}^{M} \log \boldsymbol{p}_m$ are computed over $M$ observations $\boldsymbol{p}_1, ..., \boldsymbol{p}_M$. Then, a student Drichlet Prior Network (DPN; Malinin and Gales, 2018), which models $q_{\boldsymbol{\phi}}(\boldsymbol{\pi}_x | x) = \text{Dir}(\boldsymbol{\alpha}_{\boldsymbol{\phi}})$, can be trained by minimizing the KL divergence between two Dirichlet distributions $\text{Dir}(\boldsymbol{\beta})$ and $\text{Dir}(\boldsymbol{\alpha}_{\boldsymbol{\phi}})$. While the seminal work of Malinin et al. (2020) originally minimized the forward KL divergence, $D_{\text{KL}}(\text{Dir}(\boldsymbol{\beta}) \| \text{Dir}(\boldsymbol{\alpha}_{\boldsymbol{\phi}}))$, subsequent research by Ryabinin et al.

(2021) showed that minimizing the reverse KL divergence, $D_{\text{KL}}(\text{Dir}(\boldsymbol{\alpha}_{\phi}) \ || \ \text{Dir}(\boldsymbol{\beta}))$, improves training stability. They also constrained the concentration parameters to be greater than one by $\hat{\boldsymbol{\alpha}}_{\phi} \leftarrow \boldsymbol{\alpha}_{\phi} + \mathbf{1}$ and $\hat{\boldsymbol{\beta}} \leftarrow \boldsymbol{\beta} + \mathbf{1}$, to further enhance stability. Accordingly, we adopted the loss function proposed by Ryabinin et al. (2021) to train the EnDD baseline:

$$\mathcal{L}_{\text{EnD}^2}(\phi) = \underbrace{\mathbb{E}_{\boldsymbol{q} \sim \text{Dir}(\hat{\boldsymbol{\alpha}}_{\phi})} \left[ \mathcal{H}\left(\bar{\boldsymbol{p}}, \boldsymbol{q}\right) \right]}_{\text{reconstruction term}} + \underbrace{\frac{1}{\sum_{k=1}^{K} \hat{\boldsymbol{\beta}}^{(k)}} \cdot D_{\text{KL}}\left(\text{Dir}(\hat{\boldsymbol{\alpha}}_{\phi}) \ || \ \text{Dir}(\mathbf{1})\right)}_{\text{prior term}}. \tag{39}$$

During test time, we compute the categorical probabilities as the mean of the student Dirichlet distribution, which is given by $\mathbb{E}_{\boldsymbol{q} \sim \text{Dir}(\hat{\boldsymbol{\alpha}}_{\phi})} \left[ \boldsymbol{q}^{(k)} \right] = \hat{\boldsymbol{\alpha}}_{\phi}^{(k)} / \sum_{j=1}^{K} \hat{\boldsymbol{\alpha}}_{\phi}^{(j)}$ for $k = 1, ..., K$. Notably, this is equivalent to applying the softmax operation to the DPN output, making the inference procedure identical to standard softmax-based classification neural networks.

**Functional ensemble distillation (FED; Penso et al., 2022)** aims to distill the whole distribution of predictions from the ensemble teacher, unlike previous approaches that are capable only of predicting the mean of the teacher predictions. Specifically, it proposed to minimize the maximum mean discrepancy between the predictions produced by the teachers and predictions produced by student. Enabling the student network to be a generative model that approximates the ensemble teacher distribution, FED has advantages over tasks that require covariance between predictions. As EDFM also proposes to distill ensemble teacher distribution via flow matching, FED is the most alike approach to ours among other baselines.

**Diffusion bridge network (DBN; Kim et al., 2024)** adopted I$^2$SB, an image reconstruction algorithm based on Diffusion Schrödinger Bridge to map predictions of a single teacher model to the mean of predictions of ensemble teachers. Although DBN proposed a fast ensembling algorithm based on a concurrent generative model, it fundamentally deviates from EDFM in that it does not distill the whole ensemble distribution.

