# OpenReview forum: "Ensemble Distribution Distillation via Flow Matching"
_ICML.cc/2025/Conference — ICML 2025 poster_

### Official Review · Reviewer_r2cZ · 2025-02-16

**Overall Recommendation:** 3

**Summary:**

The paper presents an ensemble distribution distillation method leveraging flow matching to efficiently transfer knowledge from an ensemble teacher to a smaller student model.

A new approach that models ensemble distribution distillation using flow matching, enabling student models to better capture the diversity of teacher ensembles.

A theoretical formulation where flow matching is applied in logit space, optimizing a vector field to map student model predictions to the ensemble teacher’s distribution.

Extensive experimental validation on image classification (CIFAR-10/100), commonsense reasoning (ARC, OBQA), and robustness evaluations under distribution shifts (CIFAR-10.1, STL, SVHN).

Demonstration that the proposed EDFM (Ensemble Distribution Flow Matching) method outperforms previous ensemble distillation approaches, improving accuracy, calibration, and efficiency while reducing computational costs.

**Claims And Evidence:**

The authors claim that flow matching enables better ensemble distribution distillation by capturing ensemble diversity more effectively than previous methods. Supported by empirical results in diversity, fidelity, and robustness experiments that show EDFM consistently outperforming baselines.

The paper claims that EDFM improves both efficiency and predictive performance compared to existing distillation techniques.
Supported by runtime analysis showing that EDFM achieves strong scalability and efficiency with minimal computational overhead.

It asserts that EDFM preserves ensemble teacher properties better than competing methods. Supported by fidelity analysis, where EDFM achieves the closest alignment with the ensemble teacher in various distance metrics (TVD, KLD, JSD).

The paper claims that flow matching naturally captures ensemble diversity, but it does not include direct theoretical justification for why flow matching should perform better in this setting compared to prior distillation methods.

**Essential References Not Discussed:**

N/A

**Experimental Designs Or Analyses:**

The experiments are comprehensive, covering different dataset shifts, evaluation metrics, and baselines.

The fidelity and diversity analyses are particularly strong, demonstrating how well the student model mimics ensemble predictions.

Ablation experiments (e.g., varying the number of flow matching steps, testing different vector field parameterizations) are missing.

The paper could include more real-world benchmarks beyond academic datasets to validate its practical effectiveness.

**Methods And Evaluation Criteria:**

The proposed flow matching approach is well-motivated and clearly formulated.

The evaluation methodology is strong, using multiple baselines (e.g., KD, EnDD, FED, DBN) and comprehensive metrics (accuracy, calibration, fidelity, efficiency).

The benchmarks (CIFAR, ARC, OBQA, STL, SVHN) are appropriate for the task.

The paper does not provide ablation studies that analyze how different design choices (e.g., type of flow matching loss, student model architecture) affect performance.

**Other Comments Or Suggestions:**

The writing could be clearer in some sections, especially in theoretical explanations (e.g., the connection between flow matching and Bayesian ensembles).

Ablation studies (e.g., varying flow matching step size, different vector field architectures) would strengthen the empirical claims.

A more explicit comparison to diffusion-based distillation methods would be useful.

**Other Strengths And Weaknesses:**

Strengths

Novelty: The paper introduces flow matching into ensemble distillation, which is a new and promising direction.

Strong empirical results: EDFM consistently outperforms existing methods across multiple tasks and metrics.

Efficiency: The method is computationally efficient, reducing the overhead of traditional ensemble models.

Comprehensive evaluation: The paper rigorously tests EDFM under distribution shifts and real-world tasks.

Weaknesses

Limited theoretical justification: The paper lacks strong formal arguments for why flow matching should improve ensemble distillation.

No ablation studies: The sensitivity of EDFM to hyperparameters is not analyzed.

Comparison to recent generative modeling approaches (e.g., score-based methods) is missing.

Some experimental design choices (e.g., why a specific type of flow matching loss is used) are not well explained.

**Questions For Authors:**

How does EDFM perform when using fewer flow matching steps? Would reducing the number of steps significantly degrade performance?

How does the choice of the vector field parameterization affect distillation quality? Have you experimented with different neural network architectures for the vector field?

How does EDFM compare to alternative generative modeling techniques for distillation? Could score-based diffusion models be used instead of flow matching?

What are the limitations of EDFM when scaling to very large student models? Does the method suffer from optimization difficulties when used with transformers?

**Relation To Broader Scientific Literature:**

The paper effectively connects ensemble distillation to generative flow matching methods.

It situates EDFM within the broader context of knowledge distillation and uncertainty estimation.

The discussion of ensemble teacher diversity is well-aligned with prior work in Bayesian deep learning.

The connection to recent advances in diffusion-based distillation (e.g., Schrödinger bridge methods) could be better explored.

The paper does not address non-Bayesian perspectives on ensemble distillation, which could be valuable for robustness.

**Theoretical Claims:**

The paper provides a theoretical formulation of flow matching for ensemble distillation.

The derivation of the vector field for flow matching in logit space is consistent with prior work on normalizing flows.

Theoretical justification for why EDFM better captures ensemble diversity is not fully developed.

The claims about exponential convergence of flow matching (inspired by prior generative modeling work) are not rigorously derived for the ensemble distillation setting.

---

> ### Author Rebuttal · Authors · 2025-04-01
>
> Thank you for your thoughtful review and for recognizing our work as a new and promising direction. We are pleased that you found our extensive experimental results valuable and especially appreciate your recognition of our fidelity and diversity analyses, which are central to our contribution. We hope our responses address any remaining concerns—please reach out if you have any further questions.
>
> > Some key comparisons to prior work (e.g., Zhang et al. 2023, Jiang 2023)...
>
> We could not find Zhang et al. (2023) and Jiang (2023) in our manuscript. Could you kindly clarify which sections you are referring to?
>
> > The connection to recent advances in diffusion-based distillation...
>
> In the context of ensemble distribution distillation, DBN (Kim et al., 2024) represents a recent advance. While we discuss its relevance and differences in Section 3 (Related Work) and Appendix B.2 (Ensemble Distillation Methods), we will further clarify this in the camera-ready version.
>
> > The paper does not address non-Bayesian perspectives...
>
> Could you kindly clarify what is meant by “non-Bayesian perspectives”? If we understand correctly, point-estimating the mean of the ensemble teacher prediction would fall under non-Bayesian approaches, while ensemble distribution distillation, which models the distribution of the ensemble teacher prediction, would be considered Bayesian. In this case, KD and DBN would correspond to the former, and EnD, FED, and our EDFM to the latter. We would appreciate any further insights.
>
> > How does EDFM perform when using fewer flow matching steps...
>
> In the runtime analysis (Figure 4, Section 5.4), we compared NFE values of {3, 5, 7} (corresponding to 3, 4, 5 steps) and selected NFE=7 for the main tables based on the performance-cost trade-off. As shown, fewer flow matching steps (NFE=3) led to slight performance drops, though NFE=5 showed no significant degradation. We agree that further ablations on the number of steps, solver, and scheduler choices would strengthen the paper.
>
> As a follow-up, we conducted an ablation study on: 1) uniform vs. exponential schedules and 2) Euler vs. Heun methods across different step counts. The figure (https://imgur.com/a/fCd15ca) summarizes the results; filled markers indicate main table settings (Heun with exponential schedule, NFE=5). This underscores the value of carefully designing the sampling procedure with performance-cost in mind—as we have already addressed. Notably, the exponential schedule, rarely used in flow matching literature, was uniquely proposed and tailored for EDFM, proving critical in our distillation setup. We will include a separate section on hyperparameter sensitivity in the camera-ready version.
>
> > How does EDFM compare to alternative generative modeling techniques...
>
> Exploring alternative generative methods is an interesting future direction. We implemented EDM (Karras et al., 2022) to train the student network and compared it with our EDFM approach. In our CIFAR-10 setup, EDM achieved an NLL of 0.224 with 35 NFEs, while EDFM reached 0.216 with only 5 NFEs; EDM was unable to achieve reasonable performance with just 5 NFEs (it achieved NLL of 0.370). It clearly shows effectiveness and efficiency of our EDFM approach.
>
> > How does the choice of the vector field parameterization...
> >
> > ...optimization difficulties when used with transformers?
>
> During the rebuttal period, we conducted an ablation study on network architectures (MLP, Transformer, and U-Net) of varying scales. The results are as follows: 1) Larger Transformers (1.31M parameters) improve NLL and ECE but are not cost-effective, as evidenced by the “#Params” and “Wall-clock time” columns (where wall-clock time refers to the duration required for 256 ensemble predictions via batched inference). 2) U-Net underperforms compared to both MLP and Transformer, likely due to its 2D spatial processing being less suited for handling logits. These findings indicate that the MLP architecture is sufficient for ensemble distribution distillation, providing a latency advantage. Furthermore, no issues related to training instability were observed, even with large student networks or Transformers.
>
> Data|Arch|# Param|ACC / NLL / ECE|Wall-clock time(microseconds)
> -|-|-|-|-
> C10|MLP|0.14M|0.929 / 0.229 / 0.013|__39__
> |||0.33M (Ours)|__0.931__ / 0.216 / 0.009|68
> |||0.70M|0.930 / 0.218 / 0.011|132
> ||Transformer|0.36M|0.930 / 0.218 / 0.011|204
> |||1.31M|__0.931__ / __0.213__ / __0.007__|724
> ||U-Net|0.44M|0.929 / 0.222 / 0.012|653
>
> > type of loss
>
> We also conducted an ablation study on the type of flow matching loss, comparing x-prediction and v-prediction, to find that the latter consistently outperforms the former. For other formulations of flow matching, please refer to our comment to review AwX6.
>
> Data|Type|ACC / NLL / ECE
> -|-|-
> C10|x-pred|__0.931__ / 0.234 / 0.024
> ||v-pred (Ours)|__0.931__ / __0.216__ / __0.009__
> C100|x-pred|0.760 / 1.031 / 0.090
> ||v-pred (Ours)|__0.761__ / __0.932__ / __0.056__

---

> > ### Comment · Reviewer_r2cZ · 2025-04-02
> >
> > Thank you for the answers. I decided to raise my score to 3.

---

> > > ### Author Response · Authors · 2025-04-02
> > >
> > > Thank you for clearly expressing your positive stance! Your detailed questions have been invaluable in shaping our additional experiments. We sincerely appreciate your constructive review once again!

---

### Official Review · Reviewer_AwX6 · 2025-03-11

**Overall Recommendation:** 4

**Summary:**

The paper presents an ensemble distillation method based on flow matching named EDFM. The core idea is to learn a mapping between Gaussian noise and the logits of a (Bayesian) teacher model conditioned on the input data. The authors first analyze the importance of diversity in the predictions of the teacher when learning the student model by incorporating several previously established methods such as Mixup augmentations and auxiliary datasets. Then, the authors examine 3 aspects of their approach compared to other distillation methods (ones that attempt to preserve the ensemble diversity and ones that don't), (1) the ability to mimic the ensemble teacher models, (2) the diversity in the students prediction, and (3) the generalization capabilities of the student. The proposed approach was examined using both image and text data.

**Claims And Evidence:**

Yes.

**Essential References Not Discussed:**

No.

**Experimental Designs Or Analyses:**

The experimental design is solid.

**Methods And Evaluation Criteria:**

Yes.

**Other Comments Or Suggestions:**

See Strengths And Weaknesses.

**Other Strengths And Weaknesses:**

Strengths:
- The paper is written in a clear and cohesive manner. The importance of ensemble diversity is first analyzed and then each aspect of the distillation goals is inspected with proper empirical evidence.
- In terms of prediction diversity EDFM outperforms all baseline methods in most cases. In addition compared to FED, the main competing approach, EDFM tends to exhibit better performance.
- The proposed approach is efficient in terms of the storage required and inference time when using a proper student network.

Weaknesses/Questions:
- In terms of novelty, if I understand correctly, the paper is a combination of previous ideas in the distillation literature (FED and DBN), namely using generative models for distillation and capturing diversity in the teacher model.
- The decision to model the logits as Gaussian random variables needs further justification. The probabilities after the Softmax layer are invariant to transformations of the logits such as shifting which is not consistent with the Gaussian assumption. Perhaps it will be more appropriate to use a different approach such CatFlow [1] which learns a flow directly in the probability space.
- In terms of accuracy, EDFM does not generalize as well as EnDD and DBN on in-distribution and distribution shift tasks. Will EDFM generalization improve by incorporating a bigger student model? A further analysis may be required here. Perhaps a different design choice for the flow model (such as the one described in the previous bullet) can help as well.
- I wonder how EDFM outperforms the teacher model by a large margin on text data. Contrasting it with image data, there the teacher model is usually better by a large margin (as also witnessed in the literature). I believe that further clarification and investigation is required here.
- Minor:
  - Line 199 left column - did you mean $z_{t}^{x}$?
  - Missing citation of rectified flow paper in line 165 right column.
  - The meaning of the abbreviation EDFM is not defined.
  - Missing reference and discussion to Table 2 in the main text.

[1] Eijkelboom, F., Bartosh, G., Andersson Naesseth, C., Welling, M., & van de Meent, J. W. (2024). Variational flow matching for graph generation. Advances in Neural Information Processing Systems, 37, 11735-11764.

**Questions For Authors:**

See Strengths And Weaknesses.

**Relation To Broader Scientific Literature:**

Relation to broader scientific literature is good.

**Theoretical Claims:**

Not relevant.

---

> ### Author Rebuttal · Authors · 2025-04-01
>
> We appreciate the positive feedback highlighting our paper’s clarity and coherence. We hope our responses address any remaining concerns, and please reach out if you have any further questions.
>
> > the paper is a combination of previous ideas in the distillation literature
>
> We respectfully differ in our interpretation of the novelty claim and would like to reference Michael Black’s remark: "The novelty, however, must be evaluated before the idea existed." To the best of our knowledge, this is the first work to introduce flow matching as a novel framework for ensemble distribution distillation. Our work is a pioneering contribution that, through extensive experiments, demonstrates how flow matching can be both efficient and effective across three key aspects—diversity, fidelity, and generalization—in the context of ensemble distribution distillation. We believe this represents a clear and novel contribution to the community.
>
> > The decision to model the logits as Gaussian random variables needs further justification.
>
> Thank you for your insightful feedback on our Gaussian modeling of logits. We, too, initially explored combining logit geometry with flow matching. However, after preliminary investigation, we found "simple is best"—alternative formulations underperformed the simplest one. Notably, prior works such as Yun et al. (2023) and Kim et al. (2024) also predict logits instead of categorical probabilities. Below, we summarize results of revisiting this comparison under the current setup:
>
> Data|Space|Distribution|ACC / NLL / ECE
> -|-|-|-
> C10|Probability simplex|SFM|0.929 / 0.244 / 0.019
> |||FM on simplex|0.929 / 0.243 / 0.015
> ||Logit space|FM w/ ILR|__0.931__ / 0.218 / 0.011
> |||FM (Ours)|__0.931__ / __0.216__ / __0.009__
>
> - __Statistical Flow Matching (SFM).__ Many flow matching formulations on the probability simplex, including CatFlow (Eijkelboom et al., 2024), are inapplicable here as they assume discrete data, whereas we handle a "continuous" logit distribution. SFM (Cheng et al., 2025) is the most relevant to our situation in this line of work, which provides a method to map the uniform distribution into arbitrary simplex distributions.
>
> - __Flow Matching on Probability Simplex (FM on simplex).__ Alternatively, one can apply flow matching directly on the simplex, mapping the uniform distribution to the ensemble distribution. This is valid as the simplex itself is a Euclidean space.
>
> - __Isometric Log-Ratio Transform (FM w/ ILR).__ As you have sharply pointed out, teacher logits are invariant under scalar shifts, yielding identical softmax outputs. Thus, forcing the student to match teacher logits “exactly” may be overly strict. The ILR transform addresses this by mapping the probability simplex of dimension $D$ into a $D-1$-dimensional vector space, avoiding redundancy.
>
> Overall, applying flow matching directly to pre-softmax logits produces better results than modeling it on post-softmax categorical probabilities. One possible explanation is that the softmax operation distorts the informative scale of the logits, negatively impacting ensemble distribution distillation. For example, unnormalized density information, calculated as the log-sum-exp of logits (Grathwohl et al., 2020), may be lost due to the softmax transformation.
>
> > Will EDFM generalization improve by incorporating a bigger student model? … Perhaps a different design choice for the flow model can help as well.
>
> During the rebuttal period, we conducted an ablation study on network architectures (MLP, Transformer, and U-Net) of varying scales (please see our comment to reviewer r2cZ for detailed results). The result demonstrates that the performance saturates with relatively small students, and the model used in the paper is already enough.
>
> > I wonder how EDFM outperforms the teacher model by a large margin on text data.
>
> This relates to two points: 1) our EDFM adopting a pre-trained teacher network, and 2) a slight decrease in accuracy for Multi-IVON compared to IVON. Specifically, for language tasks, EDFM is trained using the frozen, pre-trained IVON@mean model, which we found to outperform Multi-IVON in accuracy, consistent with Cong et al. (2024). As a result, our method improves upon the IVON@mean baseline by effectively distilling the Multi-IVON teacher (slightly in accuracy, and significantly in NLL, ECE), thereby surpassing the teacher by a wide margin in accuracy, similarly to the IVON@mean model.
>
> Method|ARC-C|ARC-E|OBQA
> -|-|-|-
> IVON@mean|__0.710__ / 1.874 / 0.246|__0.888__ / 0.645 / 0.094|__0.809__ / 0.653 / 0.111
> Multi-IVON|0.675 / __0.891__ / __0.091__|0.868 / __0.357__ / __0.026__|0.794 / __0.512__ / __0.032__
> ||
> EDFM (Ours)|0.715 / 1.113 / 0.142|0.892 / 0.390 / 0.044|0.818 / 0.537 / 0.046
>
> > Minor
>
> We sincerely appreciate your thorough review (yes, $z_{t}^{x}$ is correct); we will make sure to adjust the mistakes pointed out in the camera-ready version.

---

> > ### Comment · Reviewer_AwX6 · 2025-04-02
> >
> > Thank you for the answers. I believe the authors properly addressed my comments and those of the other reviewers; hence, I decided to raise my score to 4.

---

> > > ### Author Response · Authors · 2025-04-02
> > >
> > > We are pleased that our additional clarifications and ablation results effectively addressed the reviewers' concerns. We will incorporate them into the final manuscript in a clear and well-organized manner. Thank you again for your insightful and supportive feedback!

---

### Official Review · Reviewer_37tb · 2025-03-15

**Overall Recommendation:** 2

**Summary:**

This paper proposes a novel ensemble distribution distillation method (EDFM) that utilizes flow matching to efficiently transfer the diversity of ensembled teacher models to a smaller student model. Key challenges in ensemble distribution distillation are addressed, including the high computational cost of large integrations and the difficulty of capturing the full diversity of ensemble predictions due to the capacity constraints of student models. The proposed method introduces a lightweight network that learns to map individual model predictions to a vector field of integrated teacher prediction distributions, enabling fast and parallelizable inference. Extensive experiments demonstrate the effectiveness of EDFM compared to existing ensemble distribution distillation methods.

**Claims And Evidence:**

The claims made in the submission are supported by extensive experimental evidence, which is presented in a clear and detailed manner. The authors provide a comprehensive set of experiments across various tasks, including image classification and commonsense reasoning, to validate their proposed method, Ensemble Distribution Distillation via Flow Matching (EDFM). Here are some key points that support the claims:
1. The authors demonstrate that EDFM outperforms existing ensemble distillation methods in terms of accuracy, negative log-likelihood (NLL), and expected calibration error (ECE) across multiple datasets (CIFAR-10, CIFAR-100, ARC-C, ARC-E, OBQA).
2. The authors highlight the efficiency of EDFM in terms of runtime and computational cost. Experiment shows that EDFM scales well with the number of ensemble predictions, maintaining low execution times while improving performance.

**Essential References Not Discussed:**

[1]  Improved distribution matching distillation for fast image synthesis[J]. Advances in Neural Information Processing Systems, 2024, 37: 47455-47487.
[2]  Knowledge Distillation via Flow Matching[J].
[3]  One-step diffusion with distribution matching distillation[C]//Proceedings of the IEEE/CVF conference on computer vision and pattern recognition. 2024: 6613-6623.

**Experimental Designs Or Analyses:**

I reviewed the soundness and validity of the experimental designs and analyses presented in the submission. Overall, the experimental design is robust and well-structured, with appropriate benchmarks, metrics, and comparisons to validate the proposed method. However, the methods used in the comparisons are quite old.

**Methods And Evaluation Criteria:**

The proposed methods and evaluation criteria make sense for the problem and application at hand. The authors address the challenge of ensemble distillation, which aims to transfer the knowledge from a computationally expensive ensemble of models to a single, more efficient student model. The proposed method, Ensemble Distribution Distillation via Flow Matching (EDFM), is designed to effectively capture the diversity of ensemble predictions and transfer this knowledge to the student model.

**Other Comments Or Suggestions:**

No.

**Other Strengths And Weaknesses:**

This method presents some conceptual innovations, particularly in applying flow matching techniques to ensemble distillation tasks. However, flow matching itself is an existing technique, and the main contribution of this work lies more in integrating it with ensemble distillation rather than introducing theoretical or algorithmic breakthroughs. The authors should more clearly articulate their method's unique improvements over existing flow matching approaches (such as the design of conditional flow matching or efficient inference strategies) to better highlight its novelty.

Furthermore, the Abstract and Methods are quite simple in writing. And all compared methods are very old.

**Questions For Authors:**

NO.

**Relation To Broader Scientific Literature:**

The key contributions of the paper are deeply rooted in and extend the broader scientific literature in ensemble learning, knowledge distillation, generative modeling, uncertainty estimation, and scalability to large models. By introducing flow matching as a novel framework for ensemble distillation, the paper addresses several limitations of prior work, including the preservation of diversity, efficient sampling, and scalability to large models. The experimental results are also ok.

**Theoretical Claims:**

The submission does not contain explicit theoretical proofs that require verification. Instead, it focuses on empirical validation through extensive experiments. The authors do make several theoretical claims and provide justifications for their approach. I think it would be better to add a theoretical proof to the article.

---

> ### Author Rebuttal · Authors · 2025-04-01
>
> Thank you for the thoughtful review and clear understanding of our work, particularly in terms of diversity, efficiency, and scalability. We are glad that you appreciated our extensive experimental results and recognized our approach as both a novel framework and a conceptual innovation. We hope our responses address any remaining concerns, and please let us know if you have any further questions.
>
> > the methods used in the comparisons are quite old.
> >
> > the Abstract and Methods are quite simple in writing.
>
> We reviewed prior research on ensemble distribution distillation, from the EnDD in the seminal work of Malinin et al. (2020), which was later improved by Ryabinin et al. (2021), and the most recent FED by Penso et al. (2022) as baselines. We would greatly appreciate it if you could share further related recent methods in ensemble distribution distillation. Additionally, we will make sure to expand the abstract and method sections in the final version.
>
> > Essential References Not Discussed
>
> Thank you for sharing the relevant prior works that we may have overlooked. Shao et al., (2024) enhances knowledge distillation using flow matching, and DiffKD (Huang et al., (2023)), or KDiffusion (Yao et al., (2024)) proposed similar methods based on diffusion models. However, they are more similar to DBN (Kim et al., (2024)) rather than EDFM, as they align logits or features between teacher and student models one-to-one. Yin et al. (NeurIPS 2024, CVPR 2024) propose distribution matching distillation to distill diffusion models into one-step generators for faster sampling. As such, their goal and methodology differ from EDFM. Nonetheless, they offer valuable insights in the general context of knowledge distillation. We will include these connections in the camera-ready version.
>
> > The submission does not contain explicit theoretical proofs that require verification.
>
> As you mentioned, the main contribution of our work lies in introducing flow matching as a novel framework for ensemble distribution distillation, demonstrated through extensive experiments. Since our approach aligns with the standard flow matching scenario, we believe the theoretical claims from the existing flow matching literature apply to our work as well, without significant challenges (e.g., minimizing the conditional flow matching loss in Eq. (13) ensures the approximation of the ensemble teacher's predictive distribution $p_{1}$).
>
> > unique improvements over existing flow matching approaches (such as the design of conditional flow matching or efficient inference strategies)
>
> We believe our work is significant not only for introducing a pioneering and promising direction in ensemble distribution distillation, but also for empirically validating its effectiveness through extensive experiments, as recognized by the reviewers. Also, it demonstrates strong practicality by being highly efficient in terms of inference cost with the proposed parallelized inference strategy.
>
> To further solidify the position of our work as a pioneering study, we provide key takeaways within the ensemble distribution distillation context.  Comprehensive ablation studies on design choices in flow matching for effective ensemble distillation help clarify this aspect. In response, we conducted additional ablations, including those requested by the reviewers, which further strengthen the contribution of our work.
>
> __Sampling algorithm.__
> We conducted an ablation study on the sampling algorithm, focusing on the ODE-solver, time step scheduler, and the number of steps. (With the exponential scheduler, more steps are populated near $t=1$.) The plot (https://imgur.com/a/fCd15ca) shows the results, where the filled marker denotes the settings used in the paper.
>
> __Noise distribution.__
> We conducted an ablation study on the noise distribution during training, considering three distributions: (1) uniform, (2) exponential, and (3) beta. The choice of exponential and beta distributions was inspired by the well-known lesson that the model should learn more near the data. We found that the beta distribution achieved better results in terms of ECE.
>
> Data|Distribution|ACC / NLL / ECE
> -|-|-
> C10|Uniform|__0.931__ / 0.219 / 0.011
> ||Exp. (Ours)|__0.931__ / __0.216__ / 0.009
> ||Beta|__0.931__ / __0.216__ / __0.008__
> C100|Uniform|__0.761__ / 0.966 / 0.066
> ||Exp. (Ours)|__0.761__ / __0.932__ / 0.056
> ||Beta|0.760 / __0.932__ / __0.039__
>
> __Type and formulation of flow matching loss.__
> We ablated the flow matching loss by 1) comparing x-prediction and v-prediction (ours), and 2) comparing categorical probabilities and logits (ours). Please see our response to Reviewers AwX6 and r2cZ.
>
> __Architecture.__
> We ablated the network architectures (MLP, Transformer, and U-Net) at varying scales. We found that the current MLP architecture strikes the best cost-performance trade-off; please see our response to Reviewer r2cZ.

---

### Decision · Program_Chairs · 2025-05-01

**Decision:**

Accept (poster)

**Comment:**

The paper presents conceptual innovations, in the application of flow matching techniques to ensemble distillation, which is a new and promising direction. Strong empirical results are presented showing that EDFM consistently outperforms existing methods across multiple tasks and metrics. The method is computationally efficient, reducing the overhead of traditional ensemble models.
Through extensive experiments, the paper demonstrates how flow matching can be both efficient and effective across three key aspects—diversity, fidelity, and generalization—in the context of ensemble distribution distillation.

The claims made in the submission are supported by extensive experimental evidence, which is presented in a clear and detailed manner. The authors provide a comprehensive set of experiments across various tasks, including image classification and commonsense reasoning, to validate their proposed method. Comprehensive ablation studies are provided on design choices in flow matching for effective ensemble distillation.

The submission does not contain explicit theoretical proofs that require verification. Instead, it focuses on empirical validation through extensive experiments. The paper lacks strong formal arguments for why flow matching should improve ensemble distillation. The authors make several theoretical claims and provide justifications for their approach, but the paper could benefit from added theoretical proofs. For example, theoretical justification for why EDFM better captures ensemble diversity is not fully developed, and the claims about exponential convergence of flow matching (inspired by prior generative modeling work) are not rigorously derived for the ensemble distillation setting.